# Stick-Breaking Mixture Normalizing Flows with Component-Wise Tail Adaptation for Variational Inference

## Abstract

Normalizing flows with a Gaussian base provide a computationally efficient way to approximate posterior distributions in Bayesian inference, but they often struggle to capture complex posteriors with multimodality and heavy tails. We propose a stick-breaking mixture base with component-wise tail adaptation (StiCTAF) for posterior approximation. The method first learns a flexible mixture base to mitigate the mode-seeking bias of reverse KL divergence through a weighted average of component-wise ELBOs. It then estimates local tail indices of unnormalized densities and finally refines each mixture component using a shared backbone combined with component-specific tail transforms calibrated by the estimated indices. This design enables accurate mode coverage and anisotropic tail modeling while retaining exact density evaluation and stable optimization. Experiments on synthetic posteriors demonstrate improved tail recovery and better coverage of multiple modes compared to benchmark models. We also present a real-data analysis illustrating the practical benefits of our approach for posterior inference.

## 1 Introduction

Bayesian inference provides a principled framework for learning from data by updating prior beliefs about model parameters in light of observed evidence. For a probabilistic model with data $D$, prior distribution $p(z)$, and likelihood $p(D \mid z)$, the posterior distribution $p(z \mid D)$ follows from Bayes' theorem. In most realistic models, however, computing the exact posterior is intractable because it requires evaluating the marginal likelihood $p(D)$, which involves high-dimensional integration. Markov chain Monte Carlo (MCMC) methods yield asymptotically exact samples from the posterior but are often computationally prohibitive for large-scale or high-dimensional problems.

Variational Inference (VI) offers a scalable alternative to exact Bayesian inference by projecting the true posterior onto a tractable variational family $\mathcal{Q}$ and identifying the member $q_\phi(z)$ that is closest to $p(z \mid D)$ under a chosen divergence or distance measure. The accuracy of VI depends critically on the flexibility of the chosen variational family (Blei & Jordan, 2006). Normalizing Flows (NF) increase this flexibility by applying a sequence of invertible transformations to a simple base distribution, thereby yielding a highly expressive family while still permitting exact density evaluation (Rezende & Mohamed, 2015). NF-based VI can achieve accuracy comparable to Markov chain Monte Carlo (MCMC) methods, while maintaining the computational efficiency necessary for large-scale inference (Blei et al., 2017; Kucukelbir et al., 2017).

While NF can also be employed for density estimation–typically optimized via the forward KL divergence–posterior approximation settings differ in that direct samples from the target posterior are unavailable. In such cases, the optimization is instead performed using the reverse KL divergence $\mathrm{KL}\big(q \parallel p\big) = \mathbb{E}_{z \sim q}\big[\log q(z) - \log p(z|D)\big]$, where $q$ is obtained by applying an invertible transformation to a base distribution, such as a standard Gaussian. This objective has a well-known mode-seeking bias: it concentrates mass on a dominant mode of the posterior while ignoring other modes with smaller posterior mass. Consequently, NF-based VI may fail to capture the full multimodal structure of complex posteriors, particularly in models with well-separated or secondary modes that are important for predictive uncertainty.

Another limitation arises in representing heavy-tailed posteriors. When the base distribution is light-tailed, such as a Gaussian, the resulting variational family inherits this tail behavior regardless of the complexity of the flow transformations. Moreover, because standard flow architectures are built from Lipschitz-continuous transformations, the extent to which distances in the tail regions can be expanded is inherently bounded. This structural constraint restricts the ability to map a light-tailed base distribution into one with substantially heavier tails, making it difficult to approximate posteriors with extreme tail behavior. These tail limitations, when combined with the mode-seeking bias, can significantly impair inference quality in scenarios where both multimodality and heavy tails are present.

To address these limitations, this work makes three key contributions. First, we mitigate the mode-seeking bias of reverse KL divergence in VI by employing a stick-breaking mixture (SBM) as the variational base, which enables more faithful coverage of complex multimodal posteriors. Second, we develop a novel Monte Carlo–based estimator for the local tail index within the VI framework, providing a principled approach to adapting to heavy-tailed behavior while maintaining tractability. Third, we propose a component-wise normalizing flow architecture that combines a shared backbone with per-component Tail Transform Flows, thereby enhancing both flexibility and expressiveness. This design allows the variational posterior to accurately capture both the bulk structure and the tail behavior of the target posterior.

## 1.1 Related Works

One line of work improves multimodal distribution approximation by modifying the base distribution within normalizing flows, for example by employing a Gaussian-mixture base (Izmailov et al., 2020) or Dirichlet-process mixtures (Li et al., 2022). However, these methods are typically intended for density estimation or amortized generative modeling rather than reverse-KL posterior approximation in variational inference. In contrast, our approach is distinguished by determining mixture weights through SBM. Stick-breaking mechanisms have been applied to variational inference in other contexts, such as variational autoencoders (Nalisnick & Smyth, 2016; Joo et al., 2020), but to the best of our knowledge, they have not yet been applied to NF-based variational inference.

Additionally, to enhance robustness and expressive generalization, a parallel line of work has focused on heavy-tailed distributions in normalizing flows. Jaini et al. (2020) analyzed Lipschitz triangular flows and showed that a flow with a light-tailed Gaussian base cannot produce a heavy-tailed target; subsequently, TAF (Jaini et al., 2020), mTAF (Laszkiewicz et al., 2022), and ATAF (Liang et al., 2022) adopted a Student's-$t$ bases with varying, dimension-specific degrees of freedom to generate heavy-tailed targets. However, TAF and mTAF focus on density estimation, and while ATAF can be used for VI, it lacks a concrete initialization scheme for the degrees of freedom and underperforms on tail-index estimation. In Section 2.2, we further show that using the Cartesian product of Student's-$t$ distributions with heterogeneous degrees of freedom as the base distribution is not effective due to the autoregressive structure commonly used in normalizing flows.

There have also been attempts to address tail behavior by modifying the flow layers themselves. Hickling & Prangle (2024) propose Tail Transform Flows (TTF), a non-Lipschitz transformation designed to convert light-tailed base distributions into heavy-tailed targets. However, in the variational inference setting, no widely adopted procedure exists for estimating and initializing the tail thickness, and as a result, TTF often fail to produce genuinely heavy-tailed behavior.

## 2 Theoretical Background

### 2.1 Variational Inference with Normalizing Flows

Variational inference (VI) is a widely used technique for approximating intractable posterior distributions in Bayesian inference. Given a target posterior distribution $p(z \mid D)$, where $D$ denotes observed data and $z$ represents latent variables, VI seeks a tractable distribution $q_\phi(z)$ within a chosen variational family $\mathcal{Q}$ that closely approximates the true posterior. The expressiveness of this family is crucial in determining the quality of the approximation.

Normalizing Flows (NF) extend the flexibility of variational families by transforming a simple base distribution $q_\phi(z_0)$ into a richer distribution through an invertible and differentiable mappings $T_\psi$.

Let $\theta = (\phi, \psi)$, where $\phi$ parameterizes the base distribution and $\psi$ parameterizes the transformations. The transformed variable is defined as

$$z = T_\psi(z_0), \quad z_0 \sim q_\phi(z_0),$$

and the resulting density, obtained via the change-of-variables formula,

$$q_\theta(z) = q_\phi(T_\psi^{-1}(z)) \left| \det\left( \frac{\partial T_\psi^{-1}}{\partial z} \right) \right|,$$

is used to approximate the target posterior $p(z \mid D)$.

Since direct sampling from $p(z \mid D)$ is intractable, the parameters $\theta$ are optimized by minimizing the reverse KL divergence $\mathrm{KL}(q_\theta(z) \,\|\, p(z \mid D))$. Equivalently, this corresponds to maximizing the evidence lower bound (ELBO), defined as

$$\begin{aligned}
\mathrm{ELBO}(\theta) &= \mathbb{E}_{z \sim q_\theta} \left[ \log p(D, z) - \log q_\theta(z) \right] \\
&= \mathbb{E}_{z_0 \sim q_\phi} \left[ \log p(D, T_\psi(z_0)) - \log q_\phi(z_0) + \log \left| \det J_{T_\psi}(z_0) \right| \right],
\end{aligned} \tag{1}$$

where $J_{T_\psi}(z_0)$ denotes the Jacobian of the transformation $T_\psi$ at $z_0$.

Gradients with respect to both the base distribution parameters $\phi$ and the flow parameters $\psi$ can be efficiently estimated via Monte Carlo sampling, enabling stable and scalable optimization of the ELBO (Kingma & Welling, 2013; 2014; Rezende et al., 2014). A key limitation, however, lies in the choice of the base distribution $q_\phi(z_0)$. Most NF implementations assume a standard Gaussian base, which imposes a unimodal, light-tailed inductive bias. Even with complex flows, this restricts the capacity to approximate posteriors with well-separated modes or heavy tails. In reverse-KL settings, the problem is further compounded by the KL divergence's tendency to concentrate on dominant modes. We address this issue in Section 3 by replacing the standard Gaussian base with SBM, yielding more flexible, adaptive, and heavy-tailed approximations.

## 2.2 Heavy Tail Distributions in Normalizing Flows

To formalize the heavy-tailed behavior that motivates our design, we adopt a classification of distribution tails grounded in extreme value theory (EVT) (Bingham et al., 1989; De Haan & Ferreira, 2006). Whereas prior work on heavy-tailed normalizing flows has relied on the existence of moment-generating functions (Jaini et al., 2020) or the concentration function (Liang et al., 2022), our approach is based on regular variation. This perspective offers a unified framework that builds directly on standard EVT concepts and tools. In what follows, we introduce the definitions of tail classes;

**Definition 2.1** (Tail classes). *For $p, \alpha > 0$, define*

- $\mathcal{E}_\alpha^p := \{ X : \ \Pr(|X| \geq x) = e^{-\alpha x^p} L(x), \quad \log L(x) = o(x^p) \}$,

- $\mathcal{L}_\alpha^p := \{ X : \ \Pr(|X| \geq x) = \exp\{-\alpha(\log x)^p\} L(x), \quad \log L(x) = o\big((\log x)^p\big) \}$,

*where $L : \mathbb{R}^+ \to \mathbb{R}^+$ is a slowly varying function (i.e. $L(cx)/L(x) \to 1$, for every fixed $c > 0$). We call $\mathcal{E}_\alpha^p$ the* exponential-type (light-tailed) *class and $\mathcal{L}_\alpha^p$ the log-Weibull-type (heavy-tailed) class. Specifically, for $X \in \mathcal{L}_\alpha^1$, the exponent $\alpha$ determines the polynomial decay rate, and we refer to it as the **tail index**.*

**Definition 2.2** (Directional tail index). *For a directional vector $u$ on a unit sphere $\mathbb{S}^{d-1} \subset \mathbb{R}^d$ and a random vector $X \in \mathbb{R}^d$, if the one-sided scalar projection $[\langle u, X \rangle]_+ := \max\{\langle u, X \rangle, 0\}$ belongs to $\mathcal{L}_{\alpha_u}^1$ for some $\alpha_u \in (0, \infty)$, we define the* directional tail index *of $X$ along $u$ by $\alpha_X(u) := \alpha_u$.*

Building on these definitions, a key theoretical insight concerns the impact of Lipschitz transformations on tail behavior. The seminal work of Jaini et al. (2020) showed that normalizing flows constructed from Lipschitz triangular maps cannot transform a light-tailed Gaussian base into a heavy-tailed distribution. Later, Liang et al. (2022) generalized this result by proving that bi-Lipschitz transformations preserve tail classes, implying that a distribution cannot be mapped from light- to heavy-tailed, or vice versa. This limitation applies even to highly expressive, state-of-the-art architectures such as RealNVP and Neural Spline Flows, which are Lipschitz by construction. Within our EVT-based framework, the same conclusion holds, and the following theorem formalizes this result. The following is a restatement of the result from Liang et al. (2022) with a slight modification.

**Theorem 2.1** (Liang et al. (2022)). *Let $X$ be a random vector and let $f : \mathbb{R}^d \to \mathbb{R}^d$ be a bi-Lipschitz bijective map (i.e. $f$ and $f^{-1}$ are globally Lipschitz). If $X \in \mathcal{E}_\alpha^p$, then $f(X) \in \mathcal{E}_{\tilde{\alpha}}^p$ for some $\tilde{\alpha} > 0$. In addition, if $X \in \mathcal{L}_\alpha^p$ then $f(X) \in \mathcal{L}_\alpha^p$. In particular, no bi-Lipschitz normalizing flow can map a light-tailed base to a heavy-tailed output, vice versa.*

While Theorem 2.1 extends the impossibility result of Jaini et al. (2020), the limitation is not confined to the light–versus–heavy dichotomy. In particular, anisotropic tail-adaptive flows (ATAF) (Liang et al., 2022), which employ an anisotropic $t$-distributed base–the most flexible heavy-tailed base proposed to date–still suffer from this issue: once variables are mixed through linear layers or permutations, heterogeneous tail behaviors across dimensions cannot be faithfully preserved. As a natural corollary of the Lipschitz barrier, whenever coordinates with different tail indices interact, the effective tail index is determined by the heaviest tail among them. We formalize this observation in Theorem 2.2.

**Theorem 2.2** (Tail dominance). *Let $X = (X_1, \ldots, X_d)$ be a random vector with independent coordinates and $X_j \in \mathcal{L}_{\alpha_j}^1$ for each $1 \leq j \leq d$. Fix $i \in \{1, \ldots, d\}$ and let $Y_i = g_i(X_1, \ldots, X_i)$, where $g_i : \mathbb{R}^i \to \mathbb{R}$ is globally Lipschitz. Define the tail–influence set $S_i := \left\{ j \leq i : \exists R > 0, \, c_j > 0, \, r_0 \text{ s.t. } \max_{k \neq j} |x_k| \leq R, \, |x_j| \geq r_0 \Rightarrow |g_i(x)| \geq c_j |x_j| \right\}$. If $S_i \neq \emptyset$, then $Y_i \in \mathcal{L}_{\alpha_{Y_i}}^1$ with $\alpha_{Y_i} = \min_{j \in S_i} \alpha_j$.*

Theorem 2.2 shows that among the inputs influencing the linear-scale growth of $g_i$, the heaviest tail (i.e., the smallest $\alpha$) dominates. In practice, $g_i$ corresponds to the coordinate-wise update in autoregressive or coupling layers. In architectures such as neural spline flows, the $i$-th input $x_i$ of $g_i$ always belongs to $S_i$, so $S_i$ is guaranteed to be nonempty as long as no permutation layer precedes it. However, permutation layers (or invertible $1 \times 1$ convolutions)—commonly introduced to improve expressivity and mixing—disrupt this ordering by re-mixing inputs before they are passed into $g_i$. By Theorem 2.2, the resulting coordinate tail then collapses to the minimum among them, revealing a fundamental limitation of standard flow architectures.

## 3 PROPOSED METHOD

### 3.1 MIXTURE-BASE LEARNING

In this section, we introduce our choice of base distribution and an efficient loss-computation strategy for normalizing flows. While Gaussian mixtures have previously been employed as flow bases, to our knowledge this is the first work to use SBM. By extending finite mixtures to a fully non-parametric setting, SBM admits an unbounded number of components, with weights generated via a (generalized) stick-breaking process (see, e.g., Connor & Mosimann, 1969; Ishwaran & James, 2001):

$$q_\phi(z) = \sum_{k=1}^\infty \pi_k \, \mathcal{N}(z; \mu_k, \Sigma_k), \qquad \pi_k = v_k \prod_{j < k} (1 - v_j), \quad v_k \sim \text{Beta}(\alpha_k, \beta_k),$$

where $\phi = \{\mu_k, \Sigma_k, \alpha_k, \beta_k : k = 1, 2, \ldots\}$. This construction reduces to the standard stick-breaking process when $(\alpha_k, \beta_k) = (\alpha, \beta)$ and to the Dirichlet process when $(\alpha_k, \beta_k) = (1, \alpha)$ for all $k$. Because these choices impose a fixed monotonicity on the expected component weights, we instead employ a generalized stick-breaking mixture, which offers greater modeling flexibility. For practical implementation, we truncate the infinite mixture at $K$ components, with $K$ chosen sufficiently large.

A key challenge in estimating the ELBO in equation 1 via Monte Carlo is that differentiating through the Beta parameters $(\alpha_k, \beta_k)$ would normally require a reparameterization trick. Inspired by Roeder et al. (2017), we instead adopt an ELBO formulation that places the mixture weights $\{\pi_k\}$ outside the expectation, enabling analytic gradient computation with respect to $\alpha_k$ and $\beta_k$. The full derivation is provided in Appendix A.1:

$$\mathbb{E}_{z \sim q_\phi} f(z) = \sum_{k=1}^\infty \frac{\alpha_k}{\alpha_k + \beta_k} \left( \prod_{j < k} \frac{\beta_j}{\alpha_j + \beta_j} \right) \mathbb{E}_{z \sim q_k} f(z) \tag{2}$$

This approach eliminates the need for Kumaraswamy approximations (Kumaraswamy, 1980) for Beta draws and the Gumbel–Softmax relaxation (Jang et al., 2016) for discrete component assignments, yielding lower-variance and fully differentiable updates.

## 3.2 TAIL ESTIMATION

Estimating the tail index of a posterior distribution is challenging when only its unnormalized density is available. To address this, we propose the following simple yet effective procedure for each $k$th component. First, draw i.i.d. samples $z_1, \ldots, z_n$ from a known heavy-tailed distribution, such as a Student's-$t$ with low degrees of freedom (e.g., $\nu = 2$). For a chosen direction $\mathbf{u}$ on the unit sphere $\mathbb{S}^{d-1} \subset \mathbb{R}^d$, define the projection of each sample as $z_i^{\mathbf{u}} = z_i \mathbf{u}$. The projected samples inherit heavy-tailed behavior along $\mathbf{u}$. Ordering the projected magnitudes in decreasing order,

$$\|z_{(1)}^{\mathbf{u}}\| \geq \|z_{(2)}^{\mathbf{u}}\| \geq \cdots \geq \|z_{(n)}^{\mathbf{u}}\|,$$

and applying the estimator to the top-$j$ extremes yields

$$\hat{\xi}_{\mathbf{u}}^{(k)} = -\frac{1}{j} \sum_{i=1}^{j} \frac{\log p(\mu_k + z_{(i)}^{\mathbf{u}} \sigma_k \mid D) - \log p(\mu_k + z_{(j+1)}^{\mathbf{u}} \sigma_k \mid D)}{\log \|z_{(i)}^{\mathbf{u}}\| - \log \|z_{(j+1)}^{\mathbf{u}}\|} - 1,$$

which captures the decay rate of the distribution in direction $\mathbf{u}$. Here, $\mu_k + z_{(i)}^{\mathbf{u}} \sigma_k$ and $\mu_k + z_{(j+1)}^{\mathbf{u}} \sigma_k$ represent scaled versions of $z_{(i)}$ and $z_{(j+1)}$, adjusted for the component's location $\mu_k$ and scale $\sigma_k$. We now establish the consistency of this estimator under the following assumption.

**Assumption 3.1** (Directional Tail and Monotonicity). *For $\mu \in \mathbb{R}^d$, $\sigma > 0$, and $\mathbf{u} \in \mathbb{S}^{d-1}$, the posterior density $p(z \mid D)$ has a directional tail index $\xi_{\mathbf{u}} \in (0, \infty)$ along $\mathbf{u}$, and $p(\mu + \sigma r \mathbf{u} \mid D)$ decreases monotonically for all $r \geq r_0$, for some constant $r_0 > 0$.*

**Theorem 3.1** (Consistency of the Directional Tail-Index Estimator). *Let $z_1, \ldots, z_n \overset{\text{i.i.d.}}{\sim}$ Student's-$t_\nu$ for any $\nu > 0$. If Assumption 3.1 holds, then for any component $k$, the estimator $\hat{\xi}_{\mathbf{u}}^{(k)}$ defined above satisfies*

$$\hat{\xi}_{\mathbf{u}}^{(k)} \xrightarrow[n \to \infty]{\mathbb{P}} \xi_{\mathbf{u}}.$$

For light-tailed classes $\mathcal{E}_\alpha^p$ (e.g., Gaussian), the estimator diverges ($\hat{\xi}_{\mathbf{u}}^{(k)} \to \infty$), whereas for boundary cases heavier than any power law (e.g., $p(r) \propto (r(\log r)^\beta)^{-1}$), it converges to 0. Formal statements and proofs of these results, together with convergence-rate analyses, are provided in Appendix A.4.

## 3.3 COMPONENT-WISE TAIL TRANSFORM FLOWS

Given the optimized base distribution and the estimated tail indices, we construct a flow model that offers greater flexibility in approximation and thereby captures both the overall shape and tail behavior of the target distribution. We begin by recalling the notion of a pushforward: for a measurable map $T : \mathbb{R}^d \to \mathbb{R}^d$ and a probability measure $q$ (with a slight abuse of notation, we use $q$ to denote both a density and its induced measure), the pushforward $T_\# q$ is defined by

$$(T_\# q)(A) = q\big(T^{-1}(A)\big), \qquad A \subseteq \mathbb{R}^d \text{ measurable.}$$

Equivalently, if $Z \sim q$, then $T(Z) \sim T_\# q$.

Because the pushforward is linear over mixtures, different transforms can in principle be applied to different mixture components. In our setting, we extend this idea by introducing component-wise invertible maps $T^{(k)}$ and defining a measurable mapping on the extended space by

$$\mathcal{T}(k, x) := T^{(k)}(x), \qquad x \sim q_k.$$

The resulting distribution is

$$\mathcal{T}_\# q = \sum_{k=1}^{\infty} \pi_k \big(T_\#^{(k)} q_k\big).$$

Although this construction is no longer a single globally invertible map, it remains compatible with the requirements of normalizing flows: each $T^{(k)}$ is invertible with a tractable Jacobian determinant, exact likelihood evaluation is possible via the change-of-variables formula, and sampling can be carried out by first drawing a component index and then mapping the corresponding sample through its associated flow.

To maximize computational efficiency while still allowing flexibility in adjusting tail thickness for each component, we apply the TTF transform to the component-specific flows:

$$T_{\text{TTF}}^{(k)} = \left( T_{\text{TTF}}^{(k),(1)}, \ldots, T_{\text{TTF}}^{(k),(d)} \right) : \mathbb{R}^d \to \mathbb{R}^d,$$

with each dimension transformed as

$$T_{\text{TTF}}^{(k),(l)} \left( z; \hat{\xi}_{+e_l}^{(k)}, \hat{\xi}_{-e_l}^{(k)} \right) = \mu_k^{(l)} + \sigma_k^{(l)} \frac{s_l}{\hat{\xi}_{s_l}^{(k)}} \left[ \text{erfc} \left( \frac{|z - \mu_k^{(l)}|}{\sigma_k^{(l)} \sqrt{2}} \right)^{-\hat{\xi}_{s_l}^{(k)}} - 1 \right], \quad l = 1, \ldots, d,$$

where $e_l$ denotes the $l$-th canonical basis vector, $\mu_k^{(l)}$ and $\sigma_k^{(l)}$ are the $l$-th elements of the mean and scale of component $k$, and $s_l$ is set to $+e_l$ if $z \geq \mu_k^{(l)}$ and $-e_l$ otherwise.

This transformation is a slightly modified version of the flow proposed by Hickling & Prangle (2024), allowing distinct tail indices for the positive and negative directions in each dimension. The indices are estimated using the direction-specific procedure described in Section 3.2. The Jacobian determinant and closed-form inverse expressions for this transform are provided in Appendix A.6. Overall, the proposed variational inference framework achieves accurate approximation while preserving the tail thickness of the target distribution around each component.

**Corollary 3.1.** *Under Assumption 3.1, define the axis-wise estimators $\hat{\xi}_{\pm\mathbf{e}_l}^{(k)}$ using the directional procedure of Section 3.2, and instantiate StiCTAF with tail transforms $T_{\text{TTF}}^{(k),(l)}(\,\cdot\,; \hat{\xi}_{+\mathbf{e}_l}^{(k)}, \hat{\xi}_{-\mathbf{e}_l}^{(k)})$ for each coordinate $l \in \{1, \ldots, d\}$. Then, StiCTAF preserves the target's tail thickness in every coordinate direction.*

## 4 EXPERIMENTS

In this section, we evaluate the performance of the proposed Stick-Breaking Component-wise Tail-Adaptive Flow (StiCTAF) in two scenarios and compare it against several benchmark models. The benchmarks include flow models with a standard Gaussian base, a Gaussian mixture base, and existing heavy-tailed normalizing-flow models–TAF (Jaini et al., 2020), gTAF (Laszkiewicz et al., 2022), and ATAF (Liang et al., 2022). In addition, we consider a normalizing flow model with a stick-breaking heavy-tailed mixture base to demonstrate that a heavy-tailed mixture base alone is insufficient. All models are implemented in `PyTorch` 2.7.0+cu126 with `CUDA` 12.6 using the `normflows` library (Stimper et al., 2023), and executed on a single NVIDIA GeForce RTX 4090 GPU. Further implementation details are provided in Appendix B.

### 4.1 NORMAL-INVERSE-GAMMA DISTRIBUTION

We first consider a Normal–Inverse-Gamma (NIG) distribution whose two coordinates exhibit different tail behaviors: one light-tailed and the other heavy-tailed. This distribution frequently arises in Bayesian linear regression (BLR), with likelihood

$$y \mid \beta, \sigma^2 \sim \mathcal{N}(X\beta, \sigma^2 I_n).$$

Using the conjugate priors

$$\beta \mid \sigma^2 \sim \mathcal{N}(m_0, \sigma^2 V_0), \qquad \sigma^2 \sim \text{Inv-Gamma}(a_0, b_0),$$

the joint posterior is again Normal-Inverse-Gamma.

As a minimal two-dimensional testbed reflecting this light-versus-heavy tail structure, we adopt the product target

$$\mathcal{N}(\mu, \sigma_0^2) \times \text{Inv-Gamma}(\alpha, \beta),$$

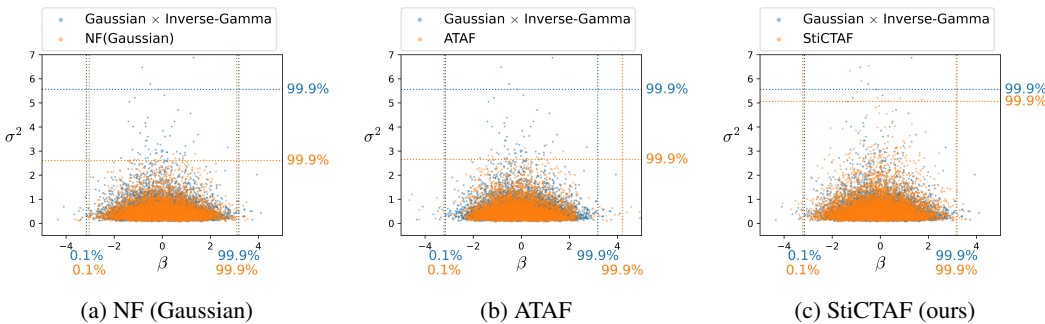

(a) NF (Gaussian)          (b) ATAF          (c) StiCTAF (ours)

Figure 1: **Normal × Inverse-Gamma Target:** Each panel compares the model and target distributions using Monte Carlo samples of size $10^4$. The dotted lines indicate the $0.1\%$ and $99.9\%$ marginal percentiles for $\beta$, and the $99.9\%$ percentile for $\sigma^2$. From left to right: NF (Gaussian), ATAF, and StiCTAF.

with parameters set to $(\mu, \sigma_0^2, \alpha, \beta) = (0.0, 1.0, 3.0, 1.0)$. In this setting, the inverse-gamma marginal along the $\sigma^2$-direction has tail index 3.0, i.e., $\Pr(\sigma^2 > t) = \Theta(t^{-3})$.

Figure 1 shows the sample percentiles for the target posterior and each NF model, with dotted guide lines indicating the $0.1\%$ and $99.9\%$ marginal percentiles for $\beta$, and the $99.9\%$ marginal percentile for $\sigma^2$. Since the target density is available in closed form, we draw $10^4$ samples from the target, using the same sample size for each NF model. The Gaussian-base NF captures the intended light tail in $\beta$ but also imposes an undesirably light tail in $\sigma$, resulting in a large discrepancy between the approximated and target $99.9\%$ lines. ATAF, even when initialized close to the oracle–$(\nu_\beta, \nu_{\sigma^2}) = (\infty, 3)$, approximated here by $(\nu_\beta, \nu_{\sigma^2}) = (30, 3)$–overestimates the upper tail in the $\beta$-direction while underestimating the extreme quantile in the $\sigma^2$-direction, deviating from the target $99.9\%$ value. In contrast, StiCTAF provides an accurate tail fit: the $\beta$-direction remains light-tailed, with extreme quantiles aligned to the target, and in the positive $\sigma^2$-direction the estimated tail index is $\hat{\xi}_{+\sigma^2} = 3.08$, very close to the target value of 3.0.

## 4.2 Complex Multimodal Target with Heavy Tails

We next test whether the proposed StiCTAF can fit a complex two-dimensional target that exhibits both heavy tails and multimodality. The target distribution is a four-component mixture: two Gaussian×Student's-$t$ components (with $\nu = 2$ and $\nu = 3$, respectively), one Two-Moons component, and one Student's-$t$ ($\nu = 2$)×Student's-$t$ ($\nu = 3$) component.

Figure 2 shows raw samples for three methods: Gaussian-base NF, Gaussian-mixture-base N , and StiCTAF. Each panel displays $2 \times 10^4$ points drawn from the target and the corresponding trained model, enabling direct comparison of how well the flows approximate the target joint distribution. The curves along the top and right margins depict the marginal densities of the horizontal and vertical coordinates.

The Gaussian-base NF fails to capture the two Gaussian×Student's-$t$ modes located on the right and at the top, concentrating mass in the lower left and center. The Gaussian-mixture-base NF recovers all modes but places excess probability mass in low-density regions, producing extraneous sample. In contrast, StiCTAF recovers all modes and more faithfully captures the tail thickness across the distribution without generating misplaced samples. Although it does not fully capture the tail of the upper Gaussian×Student's-$t$ component, it nevertheless provides the closest overall match to the target in both the bulk and the tails.

Quantitatively, since the target density is available in closed form for this synthetic experiment, we estimate the forward KL divergence $D_{\mathrm{KL}}(p\|q)$ by Monte Carlo. We also report the effective sample size (ESS), computed from importance weights $w_i = p(z_i)/q(z_i)$ with $z_i \sim q$, using

$$\mathrm{ESS} = \frac{(\sum_i w_i)^2}{\sum_i w_i^2}.$$

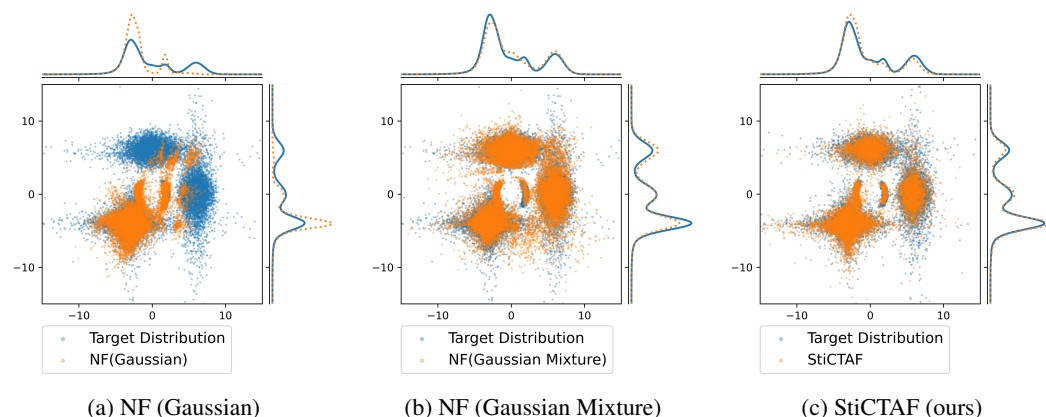

(a) NF (Gaussian)   (b) NF (Gaussian Mixture)   (c) StiCTAF (ours)

Figure 2: **Complex Multimodal Target:** Each panel compares the model and target distributions using Monte Carlo samples of size $2 \times 10^4$. The curves along the top and right margins show the univariate marginal densities. From left to right: NF (Gaussian), NF (Gaussian Mixture), and StiCTAF.

Table 1: **Forward KL-divergence and normalized ESS for Complex Mixture Target** (mean $\pm$ standard deviation)s over 10 different seeds; each repeat uses $N{=}1000$ target samples. Lower is better for KL, higher is better for ESS.

| Method | Forward KL | normalized ESS |
|---|---|---|
| NF (Gaussian) | $1.92 \pm 1.21$ | $0.31 \pm 0.17$ |
| NF (Gaussian Mixture) | $0.33 \pm 0.05$ | $0.65 \pm 0.23$ |
| StiCTAF (ours) | $\mathbf{0.22} \pm 0.09$ | $\mathbf{0.79} \pm 0.19$ |

Larger values (normalized, closer to 1) indicate better sample efficiency. Table 1 shows that StiCTAF achieves the lowest KL among all methods and the highest ESS.

## 5   REAL DATA ANALYSIS: 2024 DAILY MAXIMUM WIND SPEEDS IN KOREA

We now evaluate the performance of the proposed method on a real data application, which presents a more complicated posterior density than the simulated examples above. Specifically, we analyze daily maximum wind speed data in 2024 from the Korea Meteorological Administration (https://data.kma.go.kr/). Consecutive threshold–exceedance pairs are modeled using the logistic bivariate extreme value framework of Fawcett & Walshaw (2006). Data from four stations are considered, and we analyze four quarters of the year separately. Let $X_{(j,s),t}$ denote the daily maximum wind speed at station $j \in \{1, \ldots, 4\}$, season $s \in \{1, \ldots, 4\}$, and day $t$. For each $(j, s)$, we fix a high threshold $u_{j,s}$ and work with residuals $Y_{(j,s),t} = X_{(j,s),t} - u_{j,s}$ conditional on exceedance.

**Logistic dependence.** For $x > u$, define the exceedance–scale transform

$$Z(x) = \Lambda^{-1} \Big( 1 + \tfrac{\eta(x-u)}{\sigma} \Big)^{1/\eta}.$$

On this scale, the joint CDF for a consecutive pair of exceedances is

$$F(x_t, x_{t+1} \mid \sigma, \eta, \alpha) = 1 - \Big[ Z(x_t)^{-1/\alpha} + Z(x_{t+1})^{-1/\alpha} \Big]_+^{\alpha}, \qquad \alpha \in (0, 1],$$

where $\alpha = 1$ corresponds to independence and $\alpha \to 0^+$ to complete dependence (Fawcett & Walshaw, 2006). Full model details are provided in Appendix B.3.

**Parameterization and priors.** We decompose station and season effects using the additive models

$$\sigma_{j,s} = \mathrm{softplus}(\gamma_j^{(\sigma)}) + \mathrm{softplus}(\varepsilon_s^{(\sigma)}), \quad \eta_{j,s} = \mathrm{softplus}(\gamma_j^{(\eta)}) + \mathrm{softplus}(\varepsilon_s^{(\eta)}),$$

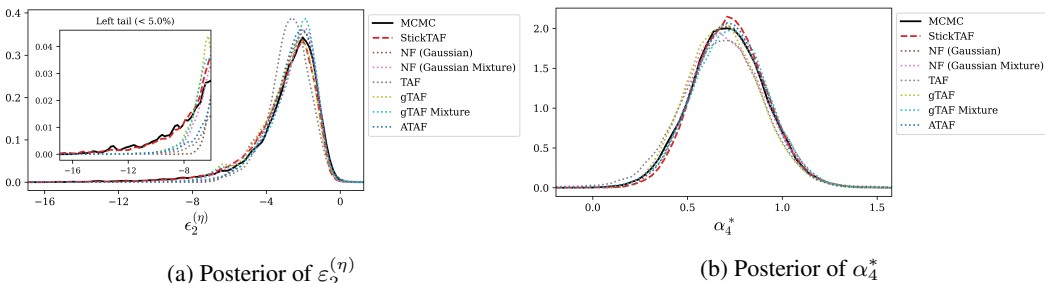

(a) Posterior of $\varepsilon_2^{(\eta)}$        (b) Posterior of $\alpha_4^*$

Figure 3: **Estimated posteriors for two parameters from the real data analysis:** Panel (a) shows $\varepsilon_2^{(\eta)}$ and panel (b) shows $\alpha_4^*$. Insets display the left 5% tail density. The black curve represents the MCMC reference, and the red curve corresponds to StiCTAF. Baselines include normalizing flows with Gaussian and Gaussian mixture bases, as well as TAF, gTAF, gTAF mixture, and ATAF.

Table 2: **Inference results for the maximum wind speed dataset.** For each parameter, the table reports the estimated mode and the 99% equal-tail credible interval. Computation times (in hours) for each method are also provided.

| Parameter | MCMC | StickTAF | NF (Gaussian) | TAF |
|---|---|---|---|---|
| $\varepsilon_1^{(\eta)}$ | -1.69 (-11.21, -0.32) | -1.81 (-11.89, -0.27) | -1.72 (-5.89, -0.31) | -1.70 (-4.71, 0.05) |
| $\varepsilon_2^{(\eta)}$ | -2.02 (-11.95, -0.38) | -2.06 (-12.09, -0.51) | -1.95 (-6.29, -0.51) | -2.60 (-7.12, 1.89) |
| $\varepsilon_3^{(\eta)}$ | -1.52 (-9.18, -0.13) | -1.62 (-10.21, -0.26) | -1.50 (-4.71, -0.22) | -1.65 (-6.05, 1.02) |
| $\varepsilon_4^{(\eta)}$ | -2.09 (-11.64, -0.50) | -2.22 (-13.88, -0.71) | -2.14 (-5.98, -0.52) | -2.32 (-5.32, 0.59) |
| Comp. time (hr) | 11.90 | 0.08 | 0.03 | 0.03 |

with station-specific $\alpha_j \in (0, 1)$. To avoid bounded supports during training, we instead infer $\alpha_j^* \in \mathbb{R}$ and set $\alpha_j = \text{sigmoid}(\alpha_j^*)$. The priors are specified as follows: $t_{\nu=10}$ for $\gamma_{1:4}^{(\sigma)}$ and $\varepsilon_{1:4}^{(\sigma)}$, $t_{\nu=3}$ for $\gamma_{1:4}^{(\eta)}$ and $\varepsilon_{1:4}^{(\eta)}$, and $\text{Beta}(1, 1)$ for $\alpha_j$.

Table 2 reports posterior modes and 99% credible intervals for selected four parameters, along with key diagnostics. MCMC is included as the gold standard, given its theoretical guarantees. As expected, all flow models are substantially faster than MCMC. Among the flow models, StiCTAF provides the tightest and most reliable 99% intervals across marginals, aligning most closely with the MCMC reference. Figure 3 displays the marginal posteriors for two representative parameters, $\varepsilon_2^{(\eta)}$ and $\alpha_4^*$. For $\varepsilon_2^{(\eta)}$, the MCMC reference shows a pronounced heavy left tail; among the approximations, only StiCTAF successfully reproduces both the tail thickness and the overall spread. For $\alpha_4^*$, which follows a light-tailed distribution, StiCTAF performs comparably to the alternatives, closely matching the central mass and dispersion. Full posterior results for all parameters are also provided in Appendix B.3.

## 6 CONCLUSION

We have introduced a variational inference framework capable of accurately representing posterior distributions that exhibit both multimodality and heavy tails. The central contribution is the replacement of a light-tailed, unimodal base distribution with SBM, whose effective number of components adapts automatically to the target distribution. To further improve tail representation, we incorporate a per-component tail transformation specifically designed to capture heavy-tailed structure when present. In both synthetic and empirical studies, the proposed method achieves close agreement with MCMC results while offering substantially greater computational efficiency. Moreover, it consistently outperforms state-of-the-art flow-based variational inference models in terms of both mode recovery and tail calibration.

ACKNOWLEDGMENTS

We acknowledge the use of large language model (LLM) tools to assist in proofreading and improving the clarity of the manuscript.

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

SUPPLEMENTARY MATERIAL FOR STICK-BREAKING MIXTURE
NORMALIZING FLOWS WITH COMPONENT-WISE TAIL ADAPTATION FOR
VARIATIONAL INFERENCE

## A  THEORETICAL DETAILS

### A.1  DERIVATION OF EQUATION 2

Here we derive the result in equation 2:

$$
\begin{aligned}
\mathbb{E}_{z \sim q_\phi} f(z) &= \int f(z) q_\phi(z)\, dz \\
&= \int f(z) \left( \sum_{k=1}^{\infty} \int_{\mathbf{v}=(v_1, v_2, \ldots)} q(z, \text{comp.} = k, \mathbf{v} \mid \mu, \Sigma, \alpha, \beta)\, d\mathbf{v} \right) dz \\
&= \int f(z) \left( \sum_{k=1}^{\infty} q_k(z) \int_{\mathbf{v}=(v_1, v_2, \ldots)} q(\text{comp.} = k \mid \mathbf{v})\, q(\mathbf{v} \mid \alpha, \beta)\, d\mathbf{v} \right) dz \\
&= \int f(z) \left( \sum_{k=1}^{\infty} q_k(z) \mathbb{E}_{\mathbf{v}} \left[ \pi_k(\mathbf{v}) \right] \right) dz \\
&= \sum_{k=1}^{\infty} \frac{\alpha_k}{\alpha_k + \beta_k} \left( \prod_{j<k} \frac{b_j}{a_j + b_j} \right) \mathbb{E}_{z \sim q_k} f(z),
\end{aligned}
$$

with $f(z) = \log p(D, z) - \log q_\theta(z)$.

### A.2  PROOF OF THEOREM 2.1

**Theorem A.1** (Liang et al. (2022)). *Let $X$ be a random vector and let $f : \mathbb{R}^d \to \mathbb{R}^d$ be a bi-Lipschitz bijective map (i.e. $f$ and $f^{-1}$ are globally Lipschitz). If $X \in \mathcal{E}_\alpha^p$, then $f(X) \in \mathcal{E}_{\tilde{\alpha}}^p$ for some $\tilde{\alpha} > 0$. In addition, if $X \in \mathcal{L}_\alpha^p$ then $f(X) \in \mathcal{L}_\alpha^p$. In particular, no bi-Lipschitz normalizing flow can map a light-tailed base to a heavy-tailed output, vice versa.*

*Proof.* Since $f$ is bi-Lipschitz, there exist $m, M > 0$ and $C \geq 0$ such that, for all $x \in \mathbb{R}^d$,

$$
m\|x\| - C \;\leq\; \|f(x)\| \;\leq\; M\|x\| + C.
$$

Hence, for all $t > 0$,

$$
\Pr\big(|X| > (t - C)/M\big) \;\geq\; \Pr\big(|f(X)| > t\big) \;\geq\; \Pr\big(|X| > (t + C)/m\big). \tag{3}
$$

(1) Assume $X \in \mathcal{E}_\alpha^p$, so $\bar{F}_X(r) = \Pr(|X| > r) = \exp\{-\alpha r^p\} L(r)$ with $L$ slowly varying and $\log L(r) = o(r^p)$. From equation 3 with $r_\pm(t) := (t \mp C)/M,\ (t \pm C)/m$ we obtain

$$
\exp\big\{ -\alpha\, r_+(t)^p \big\} L\big(r_+(t)\big) \;\geq\; \bar{F}_{f(X)}(t) \;\geq\; \exp\big\{ -\alpha\, r_-(t)^p \big\} L\big(r_-(t)\big).
$$

Since $r_\pm(t) = \Theta(t)$, we have $r_\pm(t)^p = t^p\, \theta_\pm(t)$ with $\theta_\pm(t) \to M^{-p}$ and $m^{-p}$, respectively, and $L(r_\pm(t))$ is slowly varying as a composition with an affine scaling. Therefore,

$$
\exp\{-\alpha M^{-p} t^p (1 + o(1))\}\, \tilde{L}_+(t) \;\geq\; \bar{F}_{f(X)}(t) \;\geq\; \exp\{-\alpha m^{-p} t^p (1 + o(1))\}\, \tilde{L}_-(t),
$$

for some slowly varying $\tilde{L}_\pm$. By squeezing, there exists $\tilde{\alpha} \in [\alpha/M^p,\ \alpha/m^p]$ and a slowly varying $L_f$ such that

$$
\bar{F}_{f(X)}(t) = \exp\{-\tilde{\alpha}\, t^p\} L_f(t), \qquad \log L_f(t) = o(t^p),
$$

hence $f(X) \in \mathcal{E}_{\tilde{\alpha}}^p$.

(2) Assume $X \in \mathcal{L}_\alpha^p$, so $\bar{F}_X(r) = \exp\{-\alpha(\log r)^p\} L(r)$ with $L$ slowly varying and $\log L(r) = o((\log r)^p)$. Using equation 3 again,

$$
\exp\big\{ -\alpha\big( \log r_+(t) \big)^p \big\} L\big(r_+(t)\big) \;\geq\; \bar{F}_{f(X)}(t) \;\geq\; \exp\big\{ -\alpha\big( \log r_-(t) \big)^p \big\} L\big(r_-(t)\big).
$$

Since $\log r_\pm(t) = \log t + O(1)$, we have $\big(\log r_\pm(t)\big)^p = (\log t)^p (1 + o(1))$, and $L(r_\pm(t))$ remains slowly varying as $t \to \infty$. Therefore,

$$-\log \bar{F}_{f(X)}(t) = \alpha(\log t)^p \{1 + o(1)\},$$

which is equivalent to

$$\bar{F}_{f(X)}(t) = \exp\{-\alpha(\log t)^p\} L_f(t),$$

for some slowly varying $L_f$ with $\log L_f(t) = o((\log t)^p)$. Hence $f(X) \in \mathcal{L}_\alpha^p$.

Combining (1)–(2) proves the stated closure properties. In particular, a bi-Lipschitz map cannot send a light-tailed $\mathcal{E}$-law to a heavy-tailed $\mathcal{L}$-law, nor the converse. □

## A.3 Proof of Theorem 2.2

**Theorem A.2** (Tail dominance). *Let $X = (X_1, \ldots, X_d)$ be a random vector with independent coordinates and $X_j \in \mathcal{L}_{\alpha_j}^1$ for each $1 \le j \le d$. Fix $i \in \{1, \ldots, d\}$ and let $Y_i = g_i(X_1, \ldots, X_i)$, where $g_i : \mathbb{R}^i \to \mathbb{R}$ is globally Lipschitz. Define the tail–influence set $S_i := \Big\{ j \le i : \exists R > 0, \ c_j > 0, \ r_0 \ s.t. \quad \max_{k \ne j} |x_k| \le R, \ |x_j| \ge r_0 \ \Rightarrow \ |g_i(x)| \ge c_j |x_j| \Big\}$. If $S_i \ne \emptyset$, then $Y_i \in \mathcal{L}_{\alpha_{Y_i}}^1$ with $\alpha_{Y_i} = \min_{j \in S_i} \alpha_j$.*

*Proof.* Let $i \in \{1, \ldots, d\}$ be fixed and abbreviate $Y := Y_i = g_i(X_1, \ldots, X_i)$ and $S := S_i$. Since $g_i$ is globally Lipschitz, there exist constants $L \ge 1$ and $B \ge 0$ such that

$$|g_i(x)| \ \le \ L\|x\|_1 + B \qquad (\forall x \in \mathbb{R}^i). \tag{4}$$

*Upper bound (no heavier than the heaviest input).* By equation 4 and the union bound,

$$\Pr(|Y| > t) \ \le \ \Pr\Big( \sum_{j \le i} |X_j| > (t - B)/L \Big) \ \le \ \sum_{j \le i} \Pr\big( |X_j| > (t - B)/(Li) \big).$$

For each $j$, $|X_j| \in \mathcal{L}_{\alpha_j}^1$, so $\Pr(|X_j| > x) = x^{-\alpha_j} \ell_j(x)$ with $\ell_j$ slowly varying. Hence

$$\liminf_{t \to \infty} \frac{-\log \Pr(|Y| > t)}{\log t} \ \ge \ \min_{j \le i} \alpha_j \ \ge \ \min_{j \in S} \alpha_j,$$

i.e. $\alpha_Y \ge \min_{j \in S} \alpha_j$.

*Lower bound (the heaviest influencer dominates).* Pick $j^* \in S$ with $\alpha_{j^*} = \min_{j \in S} \alpha_j$. By the definition of $S$ there exists $R > 0$ such that $|x_{j^*}| \to \infty$ with $\max_{k \ne j^*} |x_k| \le R$ implies $|g_i(x)| \to \infty$. Therefore, for each sufficiently large $t$ we can choose a threshold $b(t) > 0$ with

$$\big\{ |x_{j^*}| > b(t), \ \max_{k \ne j^*} |x_k| \le R \big\} \ \subseteq \ \{|g_i(x)| > t\}. \tag{5}$$

Specifically, Taking $b(t) = c_{j^*} t$ directly implies $|g_i(x)| > t$.

By independence,

$$\Pr(|Y| > t) \ \ge \ \Pr\big( |X_{j^*}| > b(t) \big) \cdot \Pr\Big( \max_{k \ne j^*} |X_k| \le R \Big).$$

The second factor is a positive constant $c_R \in (0, 1]$ (independent of $t$). For the first factor, $|X_{j^*}| \in \mathcal{L}_{\alpha_{j^*}}^1$ gives $\Pr(|X_{j^*}| > x) = x^{-\alpha_{j^*}} \ell_{j^*}(x)$ with $\ell_{j^*}$ slowly varying. Since $b(t) \to \infty$ as $t \to \infty$, we obtain

$$\limsup_{t \to \infty} \frac{-\log \Pr(|Y| > t)}{\log t} \ \le \ \limsup_{t \to \infty} \frac{\alpha_{j^*} \log b(t) - \log \ell_{j^*}(b(t)) - \log c_R}{\log t}.$$

Consider $b(t) = c_{j^*} t$, then $\log b(t)/\log t \to 1$ and $\log \ell_{j^*}(b(t))/\log t \to 0$ (slow variation). Therefore

$$\limsup_{t \to \infty} \frac{-\log \Pr(|Y| > t)}{\log t} \ \le \ \alpha_{j^*}.$$

Combining the upper and lower bounds shows $\alpha_Y = \min_{j \in S} \alpha_j$, and thus $Y \in \mathcal{L}_{\alpha_Y}^1$. □

### A.4 TAIL ESTIMATOR IN VARIATIONAL INFERENCE

#### A.4.1 VALIDITY OF ASSUMPTION

We restate the assumption used in the main text and propose an equivalent condition.

**Assumption A.1** (Directional Tail and Monotonicity (restated))**.** *Fix $\mu \in \mathbb{R}^d$, $\sigma > 0$, and $\mathbf{u} \in \mathbb{S}^{d-1}$. Assume the posterior density $p(z \mid D)$:*

- *$p(\cdot \mid D)$ has directional tail index $\xi_{\mathbf{u}} \in (0, \infty)$,*

- *$p(\mu + \sigma r \mathbf{u} \mid D)$ is monotonically decreasing for all $r \geq r_0$, for some constant $r_0 > 0$.*

**Lemma A.1** (Directional density regular variation)**.** *Under Assumption A.1, there exist $\xi_{\mathbf{u}} \in (0, \infty)$ and a slowly varying function $L_{\mathbf{u}} : (0, \infty) \to (0, \infty)$ such that, as $r \to \infty$,*

$$p(\mu + \sigma r \, \mathbf{u} \mid D) \;=\; r^{-(1+\xi_{\mathbf{u}})} L_{\mathbf{u}}(r) \left(1 + o(1)\right).$$

*Proof.* Fix $\mu \in \mathbb{R}^d$, $\sigma > 0$, and $\mathbf{u} \in \mathbb{S}^{d-1}$, and write $g_{\mathbf{u}}(r) = p(\mu + \sigma r \mathbf{u} \mid D)$ for $r \geq 0$. By Assumption A.1, $g_{\mathbf{u}}(r)$ is eventually monotone decreasing and the posterior has directional tail index $\xi_{\mathbf{u}} \in (0, \infty)$ along $\mathbf{u}$. Define the (one–dimensional) directional tail integral $\overline{F}_{\mathbf{u}}(r) = \int_r^\infty g_{\mathbf{u}}(s) \, \sigma \, ds$ for $r \geq 0$. The directional tail–index assumption means that $\overline{F}_{\mathbf{u}}$ is regularly varying with index $-\xi_{\mathbf{u}}$, i.e., $\overline{F}_{\mathbf{u}}(r) = r^{-\xi_{\mathbf{u}}} L_{\mathbf{u}}^{(0)}(r) \left(1 + o(1)\right)$ as $r \to \infty$, for some slowly varying $L_{\mathbf{u}}^{(0)}$.

Since $g_{\mathbf{u}}$ is eventually monotone, the monotone density theorem (Karamata theory; see Bingham et al. (1989) Th. 1.7.2) yields $g_{\mathbf{u}}(r) \sim \{\xi_{\mathbf{u}} \overline{F}_{\mathbf{u}}(r)\}/r$ as $r \to \infty$. Hence $g_{\mathbf{u}}(r) = r^{-(1+\xi_{\mathbf{u}})}\{\xi_{\mathbf{u}} L_{\mathbf{u}}^{(0)}(r)\} \left(1 + o(1)\right)$. Setting $L_{\mathbf{u}}(r) = \xi_{\mathbf{u}} L_{\mathbf{u}}^{(0)}(r)$, which is slowly varying, we obtain $p(\mu + \sigma r \, \mathbf{u} \mid D) = r^{-(1+\xi_{\mathbf{u}})} L_{\mathbf{u}}(r) \left(1 + o(1)\right)$ as $r \to \infty$, which is the claimed directional regular variation of the density along $\mathbf{u}$. $\square$

The result of Lemma A.1 is a density-level regular-variation statement along the ray $\mu + \sigma r \, \mathbf{u}$. By Karamata's theorem for integrals (De Haan & Ferreira (2006), see Th. B.1.5)

$$\Pr\big([\langle \mathbf{u}, X\rangle]_+ \geq r\big) = \int_r^\infty s^{-(1+\xi_{\mathbf{u}})} L_{\mathbf{u}}(s) \, ds \sim \frac{1}{\xi_{\mathbf{u}}} \, r^{-\xi_{\mathbf{u}}} L_{\mathbf{u}}(r),$$

so $[\langle \mathbf{u}, X\rangle]_+ \in \mathcal{L}_{\xi_{\mathbf{u}}}^1$ and, by Definition 2.2, the directional tail index satisfies $\alpha_X(\mathbf{u}) = \xi_{\mathbf{u}}$. Therefore, under the condition that $p$ is monotonically decreasing along the direction $\mathbf{u}$ over a sufficiently large range, Definition 2.2 and the results of Lemma A.1 are equivalent. Accordingly, we shall hereafter treat the two conditions interchangeably and refer to them collectively as Assumption A.1.

This additional monotonicity/regularity is mild and is satisfied by the canonical heavy-tailed families used in practice (Pareto, Student's–$t$/Cauchy, and standard scale mixtures), so in typical settings the assumption is essentially equivalent to requiring that the projection $[\langle \mathbf{u}, X\rangle]_+$ has directional tail index $\xi_{\mathbf{u}}$.

**Examples.** For a Pareto($\alpha$) distribution with threshold $x_{\min} > 0$, $\bar{F}(x) = (x_{\min}/x)^\alpha (1 + o(1))$ and $f(x) = \alpha x_{\min}^\alpha x^{-(\alpha+1)}(1 + o(1))$, so Assumption A.1 holds with $\xi_{\mathbf{u}} = \alpha$. For a Student's–$t(\nu)$ distribution (Cauchy when $\nu = 1$), the one–sided tails satisfy $\bar{F}(x) \sim C_\nu x^{-\nu}$ and $f_\nu(x) \sim C_\nu' |x|^{-(\nu+1)}$, hence the assumption holds with $\xi_{\mathbf{u}} = \nu$, absorbing any slowly varying corrections into $L_{\mathbf{u}}$.

Finally, replacing $(\mu, \sigma)$ by any fixed $(\mu_k, \sigma_k)$ only shifts $\log r$ by a constant, since $\log(\sigma_k r) = \log(\sigma r) + \log(\sigma_k/\sigma)$; this is absorbed into $L_{\mathbf{u}}$, so the asymptotic slope $-(1 + \xi_{\mathbf{u}})$ is unaffected.

#### A.4.2 PROOF OF THEOREM 3.1

**Lemma A.2.** *Let $z_1, \ldots, z_n \overset{\text{i.i.d.}}{\sim}$ Student's-$t_\nu$ with any fixed $\nu > 0$, fix a direction $\mathbf{u} \in \mathbb{S}^{d-1}$, and set $r_i := |z_i^{\mathbf{u}}| = |\langle z_i, \mathbf{u}\rangle|$. Let $r_{(1)} \geq \cdots \geq r_{(n)}$ denote the order statistics and fix integers $1 \leq i \leq j$. Define*

$$\Delta_{i,n} := \log r_{(i)} - \log r_{(j+1)}.$$

*Then*

$$\Delta_{i,n} \xrightarrow[n\to\infty]{\mathbb{P}} \frac{1}{\nu} \log\left(\frac{j+1}{i}\right).$$

*Proof. Step 1 (Tail regular variation).* Since $z_i^{\mathbf{u}}$ has a univariate $t_\nu$ law up to a positive scale, $R := |z_i^{\mathbf{u}}|$ has a regularly varying tail with index $\nu$:

$$\bar{F}(x) := \Pr(R > x) \sim C\, x^{-\nu} L(x) \qquad (x \to \infty),$$

for some slowly varying $L$ and constant $C > 0$; see, e.g., Resnick (2007, Ch. 1).

*Step 2 (Quantile representation of top order statistics).* Let $U(y) := \inf\{x : F(x) \geq 1 - \frac{1}{y}\}$ be the tail quantile function. By regular variation of $\bar{F}$, one has

$$U(y) = y^{1/\nu}\, \ell(y), \qquad y \to \infty,$$

for some slowly varying $\ell$ (Karamata theory; see Bingham et al. (1989, Thm 1.5.12)). Moreover, for fixed $m$,

$$r_{(m)} = U\left(\frac{n}{m}\right)\{1 + o_{\mathbb{P}}(1)\} \qquad (n \to \infty), \tag{6}$$

see standard order-statistic asymptotics for heavy-tailed models (e.g., Resnick (2007, Prop. 0.10, Thm. 3.3)).

*Step 3 (Log-spacing limit).* Combining equation 6 for $m = i$ and $m = j + 1$,

$$\Delta_{i,n} = \log U\left(\frac{n}{i}\right) - \log U\left(\frac{n}{j+1}\right) + o_{\mathbb{P}}(1) = \frac{1}{\nu} \log\frac{j+1}{i} + \log\frac{\ell(n/i)}{\ell(n/(j+1))} + o_{\mathbb{P}}(1).$$

Since $\ell$ is slowly varying, $\ell(n/i)/\ell(n/(j+1)) \to 1$ as $n \to \infty$ for fixed $i, j$, hence the middle term is $o(1)$. Therefore

$$\Delta_{i,n} = \frac{1}{\nu} \log\frac{j+1}{i} + o_{\mathbb{P}}(1),$$

which yields the claimed convergence in probability. Tightness follows immediately from convergence to a finite constant. $\square$

**Remark A.1.** *For comparison, if the proposal is light-tailed in the Gumbel domain (e.g., Gaussian), then $U(y) \sim \sqrt{2\log y}$ and the same calculation gives $\Delta_{i,n} \to 0$; in particular, $\Delta_{i,n}$ remains tight but shrinks to zero, reflecting slower access to the extreme region.*

We now state and prove a theorem that is slightly broader in scope than Theorem 3.1.

**Theorem A.3** (Directional consistency and tail-class behavior). *Let $z_1, \ldots, z_n \overset{\text{i.i.d.}}{\sim} \text{Student}'\text{s-}t_\nu$ for a fixed $\nu > 0$, fix $j \geq 1$ and a direction $\mathbf{u} \in \mathbb{S}^{d-1}$, and define the estimator $\hat{\xi}_{\mathbf{u}}^{(k)}$ as in Section 3.2. Under Assumption A.1,*

$$\hat{\xi}_{\mathbf{u}}^{(k)} \xrightarrow[n\to\infty]{\mathbb{P}} \xi_{\mathbf{u}}.$$

**Remark A.2** (Behavior outside the polynomial class). *The conclusions extend beyond Assumption A.1 and characterize two complementary regimes in the same density-level scale.*

Lighter than any power. *If along direction $\mathbf{u}$ the density decays faster than every polynomial,*

$$\frac{-\log p(\mu_k + \sigma_k r\, \mathbf{u} \mid D)}{\log r} \xrightarrow[r\to\infty]{} \infty,$$

*—for instance when, for some $\alpha > 0$ and slowly varying $L$,*

$$p(\mu_k + \sigma_k r\, \mathbf{u} \mid D) \sim e^{-\alpha r^p} L(r) \quad (p > 0), \qquad or \qquad p(\mu_k + \sigma_k r\, \mathbf{u} \mid D) \sim e^{-\alpha(\log r)^p} L(r) \quad (p > 1),$$

*then the slope-based estimator diverges: $\hat{\xi}_{\mathbf{u}}^{(k)} \to \infty$ in probability.*

Heavier than any power. *If along direction $\mathbf{u}$ the density decays more slowly than every polynomial,*

$$\frac{-\log p(\mu_k + \sigma_k r\, \mathbf{u} \mid D)}{\log r} \xrightarrow[r\to\infty]{} 0,$$

*—for instance when, for some slowly varying L,*

$$p(\mu_k + \sigma_k r \, \mathbf{u} \mid D) \; \sim \; e^{-\alpha(\log r)^p} L(r) \quad (0 < p < 1), \qquad \textit{or}$$

$$p(\mu_k + \sigma_k r \, \mathbf{u} \mid D) \; \sim \; r^{-1/L_0(r)}, \; \; L_0(r) \to \infty \textit{ slowly varying,}$$

*then the estimator collapses: $\hat{\xi}_{\mathbf{u}}^{(k)} \to 0$ in probability.*

*Proof of Theorem A.3.* Fix $\mathbf{u} \in \mathbb{S}^{d-1}$ and $j \geq 1$. Write $r_i := |z_i^{\mathbf{u}}|$, let $r_{(1)} \geq \cdots \geq r_{(n)}$ be the order statistics, and set $t_i := \log r_{(i)}$ and $t_0 := \log r_{(j+1)}$. Define the local difference quotients

$$Q_{i,n} \; := \; \frac{\log p(\mu_k + \sigma_k r_{(i)} \, \mathbf{u} \mid D) - \log p(\mu_k + \sigma_k r_{(j+1)} \, \mathbf{u} \mid D)}{t_i - t_0}, \qquad i = 1, \ldots, j.$$

By Assumption A.1,

$$\log p(\mu_k + \sigma_k r \, \mathbf{u} \mid D) \; = \; C_{\mathbf{u}} - (1 + \xi_{\mathbf{u}}) \log r + \ell_{\mathbf{u}}(\log r) + o(1),$$

where $\ell_{\mathbf{u}}(t) := \log L_{\mathbf{u}}(e^t)$ satisfies $\ell_{\mathbf{u}}(t + \Delta) - \ell_{\mathbf{u}}(t) \to 0$ for each fixed $\Delta > 0$. Hence

$$Q_{i,n} = -(1 + \xi_{\mathbf{u}}) + \frac{\ell_{\mathbf{u}}(t_i) - \ell_{\mathbf{u}}(t_0)}{t_i - t_0} + o(1).$$

By Lemma A.2, $t_0 \to \infty$ and $t_i - t_0 = \Delta_{i,n} \xrightarrow{\mathbb{P}} c_i > 0$ for each fixed $i \leq j$, so the fraction $(\ell_{\mathbf{u}}(t_i) - \ell_{\mathbf{u}}(t_0))/(t_i - t_0) \xrightarrow{\mathbb{P}} 0$ uniformly over $i = 1, \ldots, j$. Therefore

$$\frac{1}{j} \sum_{i=1}^{j} Q_{i,n} \xrightarrow{\mathbb{P}} -(1 + \xi_{\mathbf{u}}), \qquad \text{so} \qquad \hat{\xi}_{\mathbf{u}}^{(k)} \; = \; -\frac{1}{j} \sum_{i=1}^{j} Q_{i,n} - 1 \xrightarrow{\mathbb{P}} \xi_{\mathbf{u}}.$$

$\square$

*Proof of Remark A.2.* We show the two regimes separately. Let $t_i := \log r_{(i)}$ and $t_0 := \log r_{(j+1)}$ as above, and set

$$Q_{i,n} = \frac{\log p(\mu_k + \sigma_k e^{t_i} \mathbf{u} \mid D) - \log p(\mu_k + \sigma_k e^{t_0} \mathbf{u} \mid D)}{t_i - t_0} = -\frac{\phi(t_i) - \phi(t_0)}{t_i - t_0},$$

where $\phi(t) := -\log p(\mu_k + \sigma_k e^t \mathbf{u} \mid D)$.

*(i) Lighter than any power.* Assume $\phi(t)/t \to \infty$ as $t \to \infty$. For each fixed $i \leq j$, Lemma A.2 yields $t_i - t_0 \to c_i > 0$ in probability and $t_0 \to \infty$. Hence

$$Q_{i,n} \; = \; -\frac{\phi(t_i) - \phi(t_0)}{t_i - t_0} \xrightarrow{\mathbb{P}} -\infty,$$

so $-\frac{1}{j} \sum_{i=1}^{j} Q_{i,n} - 1 \xrightarrow{\mathbb{P}} +\infty$ and therefore $\hat{\xi}_{\mathbf{u}}^{(k)} \to \infty$ in probability.

*(ii) Heavier than any power.* Assume $\phi(t) = o(t)$ as $t \to \infty$. Then for any $\varepsilon > 0$ there exists $T$ such that $\phi(t) \leq \varepsilon t$ for all $t \geq T$. Take $t = t_0 \geq T$ and $c = t_i - t_0$; for large $n$, Lemma A.2 ensures $c \to c_i > 0$, and with $\phi$ eventually increasing we have

$$0 \; \leq \; \frac{\phi(t_0 + c) - \phi(t_0)}{c} \; \leq \; \varepsilon.$$

Thus $Q_{i,n} \xrightarrow{\mathbb{P}} 0$ for each $i \leq j$ and $j^{-1} \sum_i Q_{i,n} \xrightarrow{\mathbb{P}} 0$. Consequently, $-\frac{1}{j} \sum_{i=1}^{j} Q_{i,n} - 1 \xrightarrow{\mathbb{P}} -1$. As is standard for tail-index estimation we truncate at 0, hence $\hat{\xi}_{\mathbf{u}}^{(k)} \to 0$ in probability. $\square$

### A.4.3 CONVERGENCE RATES UNDER SECOND-ORDER REGULAR VARIATION

By making an additional assumption on the slowly varying factor $L_{\mathbf{u}}(r)$, we can compute the convergence rate of the proposed estimator $\hat{\xi}_{\mathbf{u}}^{(k)}$.

**Assumption A.2** (Second-order directional density regular variation). *In Assumption A.1, write $\ell_{\mathbf{u}}(t) := \log L_{\mathbf{u}}(e^t)$. Assume there exist an index $\rho \leq 0$ and an auxiliary function $A : (0, \infty) \to \mathbb{R}$ with $A(r) \to 0$ such that the following* uniform second-order increment *holds: for every compact set $K \subset (0, \infty)$,*

$$\sup_{x \in K} \left| \frac{\ell_{\mathbf{u}}(t + \log x) - \ell_{\mathbf{u}}(t)}{A(e^t)} - H_\rho(x) \right| \longrightarrow 0 \qquad (t \to \infty),$$

*where $H_\rho(x) = (x^\rho - 1)/\rho$ for $\rho \neq 0$ and $H_0(x) = \log x$. Equivalently, along the ray $\mu + \sigma r \mathbf{u}$ we have the* uniform log–density increment expansion

$$\sup_{x \in K} \left| \left[ \log p(\mu + \sigma r x \mathbf{u} \mid D) - \log p(\mu + \sigma r \mathbf{u} \mid D) \right] + (1 + \xi_{\mathbf{u}}) \log x - A(r) H_\rho(x) \right| = o(A(r)),$$

*as $r \to \infty$, for every compact $K \subset (0, \infty)$.*

Assumption A.2 is the standard de Haan second-order refinement for slow variation (De Haan & Ferreira, 2006). Typical examples (with $L_{\mathbf{u}}$ eventually positive) include: (i) $L_{\mathbf{u}}(r) = (\log r)^\beta$ with $\beta \in \mathbb{R}$, for which $\rho = 0$ and one may take $A(r) \asymp (\log r)^{-1}$ (hence rates in powers of $\log r$); (ii) $L_{\mathbf{u}}(r) = 1 + c\, r^\rho + o(r^\rho)$ with $\rho < 0$ and $c \neq 0$, giving $A(r) \asymp r^\rho$ (hence polynomial rates). Both examples satisfy the uniform convergence on compact $x$–sets required in Assumption A.2.

**Theorem A.4** (Rate of convergence). *Under Assumptions A.1 and A.2, with $z_i \overset{\text{i.i.d.}}{\sim} t_\nu$ ($\nu > 0$) and fixed $j \geq 1$, let $r_{(m)} := |z_{(m)}^{\mathbf{u}}|$, $t_m := \log r_{(m)}$, and denote $c_i := \lim_{n \to \infty}(t_i - t_{j+1}) = \frac{1}{\nu} \log \frac{j+1}{i}$ (Lemma A.2). Then*

$$\hat{\xi}_{\mathbf{u}}^{(k)} - \xi_{\mathbf{u}} = \kappa_{j,\nu,\rho}\, A(r_{(j+1)}) + o_{\mathbb{P}}(A(r_{(j+1)})), \qquad \kappa_{j,\nu,\rho} = \frac{1}{j} \sum_{i=1}^{j} \frac{H_\rho(e^{c_i})}{c_i}.$$

*In particular, if $A(r) \asymp (\log r)^{-\beta}$ with $\beta > 0$, then $A(r_{(j+1)}) \asymp (\log n)^{-\beta}$ and $\hat{\xi}_{\mathbf{u}}^{(k)} - \xi_{\mathbf{u}} = O_{\mathbb{P}}((\log n)^{-\beta})$; if $A(r) \asymp r^\rho$ with $\rho < 0$, then $A(r_{(j+1)}) \asymp n^{\rho/\nu}$ and $\hat{\xi}_{\mathbf{u}}^{(k)} - \xi_{\mathbf{u}} = O_{\mathbb{P}}(n^{\rho/\nu})$.*

*Proof of Theorem A.4.* Fix $\mathbf{u} \in \mathbb{S}^{d-1}$ and $j \geq 1$. Let $r_{(m)} := |z_{(m)}^{\mathbf{u}}|$, $t_m := \log r_{(m)}$ and

$$Q_{i,n} := \frac{\log p(\mu_k + \sigma_k r_{(i)} \mathbf{u} \mid D) - \log p(\mu_k + \sigma_k r_{(j+1)} \mathbf{u} \mid D)}{t_i - t_{j+1}}, \qquad i = 1, \ldots, j.$$

By Assumption 3.1, along the ray we can write

$$\log p(\mu_k + \sigma_k e^t \mathbf{u} \mid D) = C_{\mathbf{u}} - (1 + \xi_{\mathbf{u}}) t + \ell_{\mathbf{u}}(t), \quad \text{with} \quad \ell_{\mathbf{u}}(t) := \log L_{\mathbf{u}}(e^t).$$

Hence

$$Q_{i,n} = -(1 + \xi_{\mathbf{u}}) + \frac{\ell_{\mathbf{u}}(t_i) - \ell_{\mathbf{u}}(t_{j+1})}{t_i - t_{j+1}}.$$

By Lemma A.2, $t_{j+1} \to \infty$ and $t_i - t_{j+1} \overset{\mathbb{P}}{\to} c_i > 0$ with $c_i = \frac{1}{\nu} \log \frac{j+1}{i}$. Set $x_{i,n} := \exp(t_i - t_{j+1})$. Then $x_{i,n} \overset{\mathbb{P}}{\to} e^{c_i}$ and, for all large $n$, the random multipliers $x_{i,n}$ take values in a common compact set $K \subset (0, \infty)$ (since $\{c_i\}_{i=1}^{j}$ is finite). Apply Assumption A.2 with $t = t_{j+1}$ and $x = x_{i,n}$ to obtain the *uniform second-order increment*

$$\ell_{\mathbf{u}}(t_i) - \ell_{\mathbf{u}}(t_{j+1}) = A(e^{t_{j+1}}) H_\rho(x_{i,n}) + o(A(e^{t_{j+1}}))$$

uniformly over $i = 1, \ldots, j$. Therefore,

$$Q_{i,n} = -(1 + \xi_{\mathbf{u}}) + \frac{A(e^{t_{j+1}})}{t_i - t_{j+1}} H_\rho(x_{i,n}) + o_{\mathbb{P}}(A(e^{t_{j+1}})).$$

Averaging over $i = 1, \dots, j$ and using the continuous mapping theorem with $t_i - t_{j+1} \xrightarrow{\mathbb{P}} c_i$ and $x_{i,n} \xrightarrow{\mathbb{P}} e^{c_i}$ yields

$$\frac{1}{j} \sum_{i=1}^{j} Q_{i,n} = -(1 + \xi_{\mathbf{u}}) + \left( \frac{1}{j} \sum_{i=1}^{j} \frac{H_\rho(e^{c_i})}{c_i} \right) A(e^{t_{j+1}}) + o_{\mathbb{P}}\big(A(e^{t_{j+1}})\big).$$

By the definition of the estimator,

$$\hat{\xi}_{\mathbf{u}}^{(k)} = -\frac{1}{j} \sum_{i=1}^{j} Q_{i,n} - 1 = \kappa_{j,\nu,\rho} \, A(e^{t_{j+1}}) + o_{\mathbb{P}}\big(A(e^{t_{j+1}})\big),$$

with $\kappa_{j,\nu,\rho} = \frac{1}{j} \sum_{i=1}^{j} \frac{H_\rho(e^{c_i})}{c_i}$. Finally, $e^{t_{j+1}} = r_{(j+1)}$ gives the stated expansion.

For the concrete rates, recall that for $t_\nu$ proposals the $(j+1)$-st upper order statistic satisfies $r_{(j+1)} \asymp n^{1/\nu}$, so that: (i) if $A(r) \asymp (\log r)^{-\beta}$ with $\beta > 0$, then $A(r_{(j+1)}) \asymp (\log n)^{-\beta}$; (ii) if $A(r) \asymp r^\rho$ with $\rho < 0$, then $A(r_{(j+1)}) \asymp n^{\rho/\nu}$. This yields the two $O_{\mathbb{P}}(\cdot)$ displays and completes the proof. $\square$

## A.5 COMPONENT-WISE PUSHFORWARD

**Construction of a probability measure.** Let $(\pi_k)_{k \in \mathbb{N}}$ be a probability mass function on $\mathbb{N}$ (i.e., $\pi_k \geq 0$ and $\sum_{k=1}^{\infty} \pi_k = 1$). For each $k \in \mathbb{N}$, let $q_k$ be a probability measure on $(\mathbb{R}^d, \mathcal{B}(\mathbb{R}^d))$ and let $T^{(k)} : \mathbb{R}^d \to \mathbb{R}^d$ be a measurable map. Define $\nu : \mathcal{B}(\mathbb{R}^d) \to [0, 1]$ by

$$\nu(A) := \sum_{k=1}^{\infty} \pi_k \, q_k\big((T^{(k)})^{-1}(A)\big), \qquad A \in \mathcal{B}(\mathbb{R}^d).$$

Then $\nu$ is a probability measure on $(\mathbb{R}^d, \mathcal{B}(\mathbb{R}^d))$. *Proof.* For each fixed $k$, the set function $A \mapsto q_k\big((T^{(k)})^{-1}(A)\big)$ is a measure because $T^{(k)}$ is measurable and $q_k$ is a measure. Since $\pi_k \geq 0$, Tonelli's theorem implies that the pointwise countable sum of measures is a measure; thus $\nu$ is countably additive and $\nu(\varnothing) = 0$. Moreover,

$$\nu(\mathbb{R}^d) = \sum_{k=1}^{\infty} \pi_k \, q_k\big((T^{(k)})^{-1}(\mathbb{R}^d)\big) = \sum_{k=1}^{\infty} \pi_k \, q_k(\mathbb{R}^d) = \sum_{k=1}^{\infty} \pi_k = 1,$$

so $\nu$ has total mass one. Hence $\nu$ is a probability measure. $\square$

*Equivalent product-space construction.* On the product space $(\mathbb{N} \times \mathbb{R}^d, \, 2^{\mathbb{N}} \otimes \mathcal{B}(\mathbb{R}^d))$, define the probability measure $\mu$ by

$$\mu(\{k\} \times A) := \pi_k \, q_k(A), \qquad k \in \mathbb{N}, \ A \in \mathcal{B}(\mathbb{R}^d),$$

and the measurable map $\mathcal{T} : \mathbb{N} \times \mathbb{R}^d \to \mathbb{R}^d$ by $\mathcal{T}(k, x) := T^{(k)}(x)$. Then $\mathcal{T}_{\#}\mu$ is a probability measure on $\mathbb{R}^d$ and satisfies

$$(\mathcal{T}_{\#}\mu)(A) = \sum_{k=1}^{\infty} \pi_k \, q_k\big((T^{(k)})^{-1}(A)\big) = \nu(A),$$

so the component-wise transform followed by marginalization over the index yields the same $\nu$.

**Change-of-variables for component-wise transforms and the resulting density.** Assume each $q_k$ admits a density (again denoted $q_k$) with respect to Lebesgue measure and each $T^{(k)} : \mathbb{R}^d \to \mathbb{R}^d$ is a bijective $C^1$ map with measurable inverse and Jacobian $J^{(k)}(x) = \nabla T^{(k)}(x)$. For $z \in \mathbb{R}^d$, write $x_k := (T^{(k)})^{-1}(z)$. The change-of-variables formula gives

$$(T_{\#}^{(k)} q_k)(z) = q_k(x_k) \big| \det J_{(T^{(k)})^{-1}}(z) \big| = q_k(x_k) \big| \det J^{(k)}(x_k) \big|^{-1}.$$

Consequently, the density of the (countably) infinite transformed mixture $\nu = \sum_{k=1}^{\infty} \pi_k \, T_{\#}^{(k)} q_k$ is

$$q(z) = \sum_{k=1}^{\infty} \pi_k \, q_k\big((T^{(k)})^{-1}(z)\big) \big| \det J^{(k)}\big((T^{(k)})^{-1}(z)\big) \big|^{-1}, \qquad z \in \mathbb{R}^d,$$

whenever the sum is finite (e.g., by Tonelli/DCT under mild tail and Jacobian growth controls). This identity is sufficient for exact likelihood evaluation of transformed mixtures.

**Sampling and likelihood evaluation (combined).** Sampling follows directly from the product-space viewpoint in A.X.1: draw $K \sim \text{Categorical}(\pi_1, \pi_2, \dots)$, then $X \sim q_K$, and output $Z = T^{(K)}(X)$, i.e., $Z = \mathcal{T}(K, X) \sim \mathcal{T}_{\#}\mu = \nu$. For likelihood evaluation at $z$, compute for each $k$ the inverse $x_k = (T^{(k)})^{-1}(z)$ and $\log|\det J^{(k)}(x_k)|$, then assemble

$$\log q(z) = \log \sum_{k=1}^{\infty} \exp\Big\{ \log \pi_k + \log q_k(x_k) - \log|\det J^{(k)}(x_k)| \Big\},$$

using standard log-sum-exp stabilization.

A.6 TAIL TRANSFORM FLOWS

This section presents further details on Tail Transform Flows (Hickling & Prangle, 2024), which we employ to perform component-wise transformations from light-tailed base distributions to heavy-tailed targets.

For component $k$ and coordinate $l$, write

$$x^{(l)} \;=\; T_{\text{TTF}}^{(k),(l)}\Big(z^{(l)}; \hat{\xi}_{+e_l}^{(k)}, \hat{\xi}_{-e_l}^{(k)}\Big) = \mu_k^{(l)} + \sigma_k^{(l)} \frac{s_l}{\hat{\xi}_{s_l}^{(k)}} \left[ \text{erfc}\Big( \frac{|z^{(l)} - \mu_k^{(l)}|}{\sigma_k^{(l)}\sqrt{2}} \Big)^{-\hat{\xi}_{s_l}^{(k)}} - 1 \right],$$

where $s_l = \text{sign}(z^{(l)} - \mu_k^{(l)}) \in \{+1, -1\}$ selects the right/left tail and we denote

$$r^{(l)} \;=\; \frac{z^{(l)} - \mu_k^{(l)}}{\sigma_k^{(l)}}, \qquad u^{(l)} \;=\; \frac{|r^{(l)}|}{\sqrt{2}}.$$

**Forward Jacobian.** Differentiating the scalar map in $z^{(l)}$ yields a closed form:

$$\frac{\partial T_{\text{TTF}}^{(k),(l)}}{\partial z^{(l)}}(z^{(l)}) \;=\; \frac{\sqrt{2}}{\sqrt{\pi}} \exp\Big( -\frac{(r^{(l)})^2}{2} \Big) \text{erfc}\Big( \frac{|r^{(l)}|}{\sqrt{2}} \Big)^{-(\hat{\xi}_{s_l}^{(k)}+1)}.$$

Hence the full Jacobian determinant of the component-wise transform $T_{\text{TTF}}^{(k)} = \big(T_{\text{TTF}}^{(k),(1)}, \dots, T_{\text{TTF}}^{(k),(d)}\big)$ is the product

$$\Big| \det J_{T_{\text{TTF}}^{(k)}}(z) \Big| = \prod_{l=1}^{d} \frac{\sqrt{2}}{\sqrt{\pi}} \exp\Big( -\frac{1}{2}\big( \frac{z^{(l)}-\mu_k^{(l)}}{\sigma_k^{(l)}} \big)^2 \Big) \text{erfc}\Big( \frac{|z^{(l)} - \mu_k^{(l)}|}{\sigma_k^{(l)}\sqrt{2}} \Big)^{-(\hat{\xi}_{s_l}^{(k)}+1)}.$$

*Monotonicity and smoothness.* Each scalar map is strictly increasing (derivative $> 0$), and it is $C^1$ at $z^{(l)} = \mu_k^{(l)}$ with $\partial T/\partial z^{(l)}\big|_{z^{(l)}=\mu_k^{(l)}} = \sqrt{2/\pi}$, independent of $\hat{\xi}$ and $\sigma_k^{(l)}$.

**Closed-form inverse.** Given $x^{(l)}$, define

$$t^{(l)} \;=\; \frac{x^{(l)} - \mu_k^{(l)}}{\sigma_k^{(l)}}, \qquad s_l \;=\; \text{sign}(t^{(l)}), \qquad E^{(l)} \;=\; \Big( 1 + s_l \hat{\xi}_{s_l}^{(k)} t^{(l)} \Big)^{-1/\hat{\xi}_{s_l}^{(k)}}.$$

Then the inverse map is

$$\big(T_{\text{TTF}}^{(k),(l)}\big)^{-1}(x^{(l)}) \;=\; \mu_k^{(l)} + \sigma_k^{(l)} s_l \sqrt{2} \, \text{erfc}^{-1}\big(E^{(l)}\big).$$

**Inverse Jacobian.** Let $u_*^{(l)} = \text{erfc}^{-1}\big(E^{(l)}\big)$. The per-coordinate inverse derivative is

$$\frac{\partial (T_{\text{TTF}}^{(k),(l)})^{-1}}{\partial x^{(l)}}(x^{(l)}) \;=\; \frac{\sqrt{\pi}}{\sqrt{2}} \, \text{erfc}\big(u_+^{(l)}\big)^{\hat{\xi}_{s_l}^{(k)}+1} \exp\Big( \big(u_*^{(l)}\big)^2 \Big).$$

Consequently,

$$\Big| \det J_{(T_{\text{TTF}}^{(k)})^{-1}}(x) \Big| = \prod_{l=1}^{d} \frac{\sqrt{\pi}}{\sqrt{2}} \, \text{erfc}\big(u_*^{(l)}\big)^{\hat{\xi}_{s_l}^{(k)}+1} \exp\Big( \big(u_*^{(l)}\big)^2 \Big), \quad u_*^{(l)} = \text{erfc}^{-1}\Big( \big(1 + s_l\hat{\xi}_{s_l}^{(k)}t^{(l)}\big)^{-1/\hat{\xi}_{s_l}^{(k)}} \Big).$$

Table 3: **Estimated percentiles of Gaussian-Inverse-Gamma distribution** Reported are the 0.1- and 99.9-percentiles for the first coordinate and the 99.9-percentile for the second coordinate from sample of 10,000.

| Method | $\beta$ 0.1-ptile | $\beta$ 99.9-ptile | $\sigma^2$ 99.9-ptile |
|---|---|---|---|
| Target Gaussian-Inverse-Gamma | $-3.15$ | 3.18 | 5.56 |
| NF (Gaussian) | $-3.06$ | 3.01 | 2.63 |
| NF (Gaussian Mixture) | $-2.91$ | 3.08 | 3.30 |
| TAF | $-7.98$ | 8.48 | 5.86 |
| gTAF | $-2.98$ | 3.06 | 3.07 |
| gTAF Mixture | $-3.21$ | 3.20 | 2.71 |
| ATAF | $-3.17$ | 4.33 | 2.81 |
| StiCTAF | $-3.11$ | 3.11 | 5.15 |

## B  EXPERIMENTS DETAILS

This section provides further details for the experimental results conducted on Section 4 and Section 5. As previously illustrated, we include flow models with a standard Gaussian base, a Gaussian mixture base, TAF (Jaini et al., 2020), gTAF (Laszkiewicz et al., 2022), and ATAF (Liang et al., 2022) as the benchmark methods. In addition, we consider a normalizing flow model with a stick–breaking heavy–tailed mixture base to show that a heavy–tailed mixture base alone is insufficient. See Section C for the details.

For every flow based models, we use $N = 2$ blocks, each block consisting of an autoregressive rational–quadratic spline (ARQS) transform with 3 bins and a coupling network with 64 hidden units, followed by a learnable LU-linear permutation layer. For every mixture base flow models including StiCTAF, $K = 20$ mixture components are used. For the numerical stability, the TTF were applied to those components with expected weight higher than $1 \times 10^{-2}$. All models are implemented in `PyTorch` 2.7.0+cu126 with `CUDA` 12.6 using the `normflows` library (Stimper et al., 2023), and are executed on a single NVIDIA GeForce RTX 4090 GPU.

### B.1  GAUSSIAN-INVERSE-GAMMA DISTRIBUTION

This section details the numerical experiment in Section 4, where the target distribution is

$$(\beta, \sigma^2) \sim \mathcal{N}(0, 1) \times \text{Inv-Gamma}(\text{shape} = 3, \text{scale} = 1).$$

We ran 300 iterations for ATAF and for NF with Gaussian and Gaussian–mixture bases; 500 iterations for TAF and StiCTAF (with 450 for base learning and 50 for flow learning); and 1000 iterations for gTAF and gTAF–Mixture, until convergence. The base distributions were initialized as Student's $t$ distributions with degrees of freedom 4 for TAF, $(30, 3)$ for ATAF, $(30, 2.89)$ for gTAF, and $(30, 2.72)$ for gTAF–Mixture. For the last two, the initial degrees of freedom were estimated using the tail estimator proposed in Section 3.2. Student's-$t$-based methods showed highly unstable training process, so the initial degrees of freedom where clamped greater than 2.0, and 4.0 for TAF which showed the highest instability. All method used learning rate of $5 \times 10^{-3}$.

Figure 4 presents the results across all methods, based on $10^4$ samples from each approximating distribution. The Dotted reference lines indicate the 0.1% and 99.9% marginal percentiles for $\beta$, and the 99.9% marginal percentile for $\sigma^2$. Table 3 summarizes these values for each method. We observe for TAF and ATAF that mixing across dimensions leads to a failure to accurately capture the tail thickness of $\beta$, which corroborates Theorem 2.2. Only StiCTAF accurately captures the tail behavior in both $\beta$ and $\sigma^2$.

### B.2  COMPLEX MIXTURE TARGET DISTRIBUTION

For the second numerical experiment in Section 4, the target distribution is a mixture of four components: two Gaussian $\times$ Student-$t$ components (with $\nu = 2$ and $\nu = 3$, respectively), one Two-Moons component, and one Student-$t(\nu = 2) \times$ Student-$t(\nu = 3)$ component. The component centers are $(6, 0)$, $(0, 6)$, $(-3, -4)$, and $(0, 0)$, and the mixture weights are $(0.2, 0.2, 0.1, 0.5)$ in the same order.

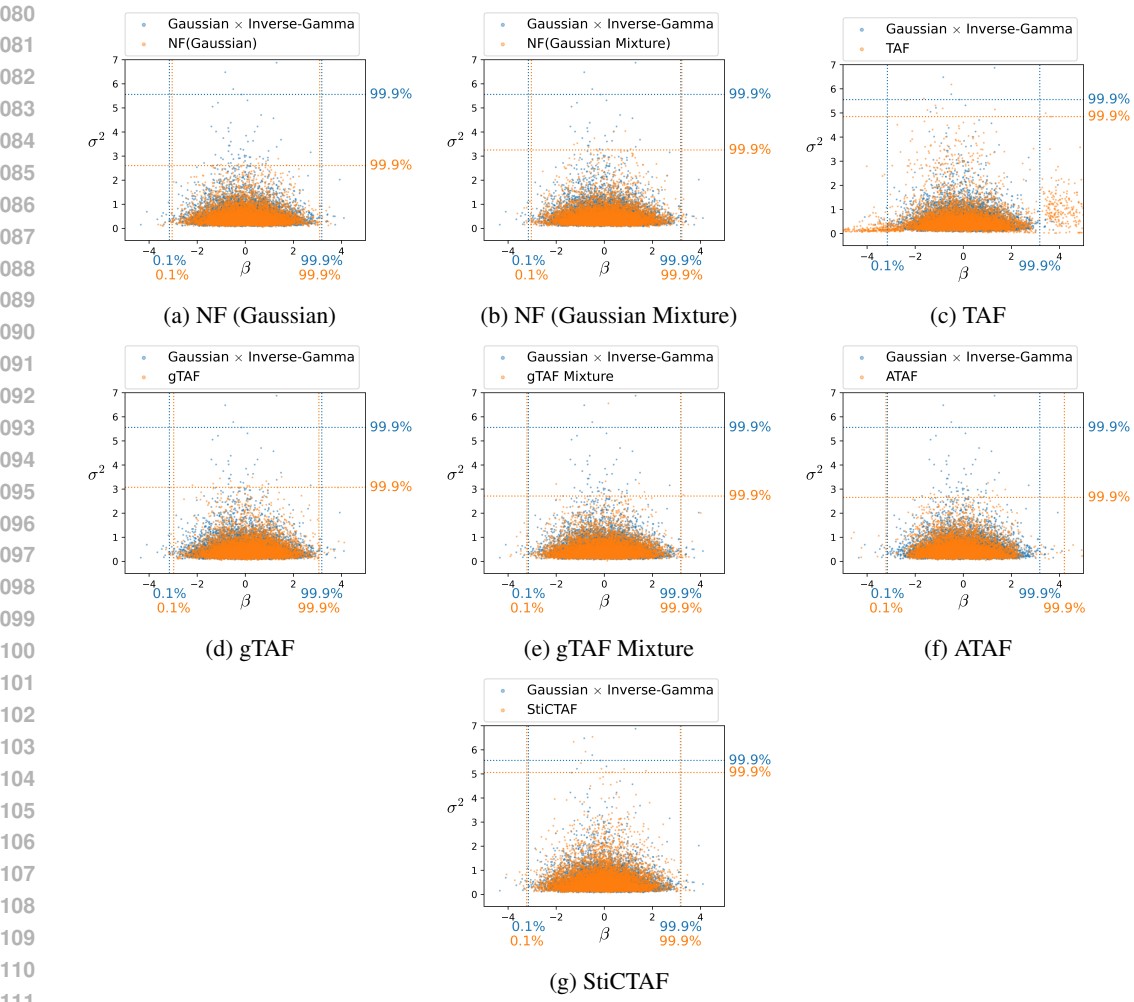

(a) NF (Gaussian)  (b) NF (Gaussian Mixture)  (c) TAF

(d) gTAF  (e) gTAF Mixture  (f) ATAF

(g) StiCTAF

Figure 4: **Normal × Inverse-Gamma Target:** Full comparison with benchmark methods using samples of size $10^4$ per model; dotted lines indicate the $0.1\%/99.9\%$ marginal percentiles.

We ran $1500$ iterations for NF (Gaussian Mixture) and $1000$ iterations for all other methods (for StiCTAF: $800$ iterations for base learning and $200$ for flow learning), continuing until convergence. The initial degrees of freedom were $(2, 2)$ for TAF, ATAF, and gTAF–Mixture, and $(2.54, 2)$ for gTAF. All methods used a learning rate of $1 \times 10^{-3}$.

Figure 5 shows $2 \times 10^4$ samples from each approximating distribution, with marginal density curves displayed along the top and right margins. It is evident that mixture-based flow models (Gaussian Mixture, gTAF–Mixture, and StiCTAF) more effectively recover all modes compared with unimodal-based flow models. For a quantitative comparison, Table 4 summarizes the mean and standard deviation of the forward Kullback–Leibler (KL) divergence (estimated via Monte Carlo using true target samples) and the normalized effective sample size (ESS) computed from samples drawn from the trained models. Among all methods, StiCTAF attains the best performance—yielding the lowest KL divergence and the highest ESS.

### B.3 REAL DATA ANALYSIS

We analyze the daily maximum wind speed data for the year 2024 obtained from the Korea Meteorological Administration (https://data.kma.go.kr/). We consider four stations ($J = 4$) and four seasons ($S = 4$). Let $X_{(j,s),t}$ denote the daily maximum wind at station $j \in \{1, \ldots, J\}$, season $s \in \{1, \ldots, S\}$, and day index $t$. For each $(j, s)$, we fix a high threshold $u_{j,s}$ and work with

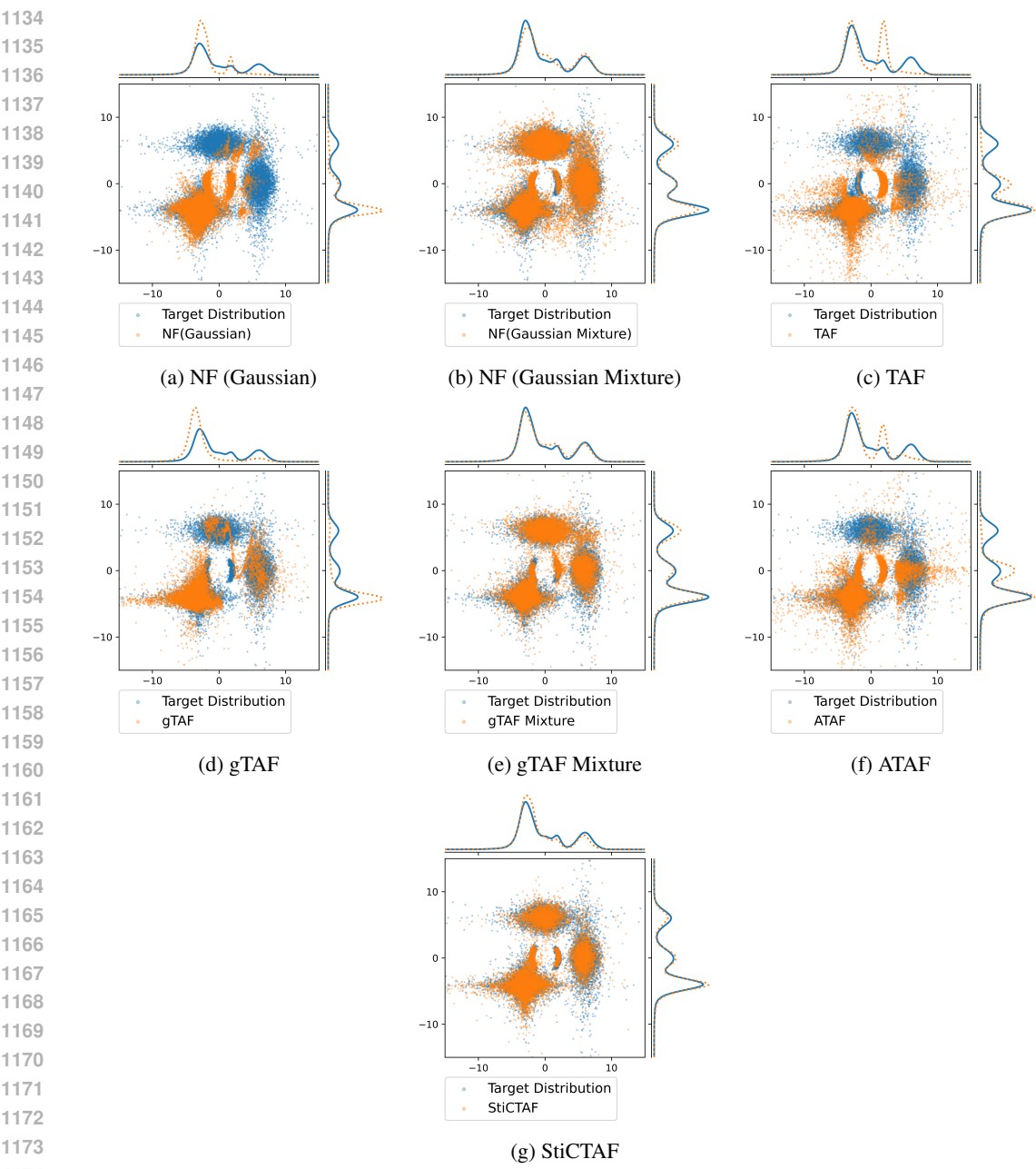

Figure 5: **Complex Multimodal Target:** Full comparison with benchmark methods using samples of size $10^4$ per model; curves along the top and right margins show the marginal densities.

exceedance residuals

$$Y_{(j,s),t} = X_{(j,s),t} - u_{j,s} \quad \text{conditioned on } X_{(j,s),t} > u_{j,s}.$$

Following Fawcett & Walshaw (2006), we adopt a pairwise extreme-value framework: threshold exceedances are modeled with GPD margins, and dependence between consecutive days is captured by a logistic bivariate extreme-value model. We use this structure to analyze the 2024 KMA wind data across stations and seasons.

**GPD marginal for threshold exceedances.** Following Fawcett & Walshaw (2006), conditional on exceeding $u_{j,s}$, the residual $Y_{(j,s),t} = X_{(j,s),t} - u_{j,s}$ is modeled by a Generalized Pareto Distribution

Table 4: **Forward KL-divergence and normalized ESS** Scores for estimating complex multimodal target distributions. (mean $\pm$ standard deviation)s over 10 random seeds are reported. Lower KL and higher ESS is the better.

| Method | Forward KL | ESS(normalized) |
|---|---|---|
| NF (Gaussian) | $1.92 \pm 1.21$ | $0.31 \pm 0.17$ |
| NF (Gaussian Mixture) | $0.33 \pm 0.05$ | $0.65 \pm 0.23$ |
| TAF | $0.90 \pm 0.09$ | $0.21 \pm 0.06$ |
| gTAF | $2.80 \pm 0.20$ | $0.07 \pm 0.07$ |
| gTAF Mixture | $0.43 \pm 0.12$ | $0.48 \pm 0.20$ |
| ATAF | $0.94 \pm 0.33$ | $0.19 \pm 0.07$ |
| StiCTAF | $\mathbf{0.22} \pm 0.09$ | $\mathbf{0.79} \pm 0.19$ |

(GPD) with scale parameter $\sigma_{j,s} > 0$ and shape parameter $\eta_{j,s}$. Its CDF and PDF are

$$H(y \mid \sigma_{j,s}, \eta_{j,s}; u_{j,s}) = 1 - \left( 1 + \frac{\eta_{j,s} y}{\sigma_{j,s}} \right)^{-1/\eta_{j,s}}, \qquad y \in \mathcal{S}(\sigma_{j,s}, \eta_{j,s}), \qquad (7)$$

$$h(y \mid \sigma_{j,s}, \eta_{j,s}; u_{j,s}) = \frac{1}{\sigma_{j,s}} \left( 1 + \frac{\eta_{j,s} y}{\sigma_{j,s}} \right)^{-1/\eta_{j,s}-1}, \qquad y \in \mathcal{S}(\sigma_{j,s}, \eta_{j,s}) \qquad (8)$$

respectively, where the support is

$$\mathcal{S}(\sigma_{j,s}, \eta_{j,s}) = \{ y \geq 0 : 1 + \eta_{j,s} y / \sigma_{j,s} > 0 \} = \begin{cases} [0, \infty) & \text{if } \eta_{j,s} \geq 0, \\ [0, -\sigma_{j,s}/\eta_{j,s}) & \text{if } \eta_{j,s} < 0. \end{cases}$$

In the limit $\eta_{j,s} \to 0$, equation 7 reduces to the exponential CDF $H(y) = 1 - \exp(-y/\sigma_{j,s})$. When $\eta_{j,s} > 0$, the GPD is heavy-tailed with Pareto-type tail index $1/\eta_{j,s}$; when $\eta_{j,s} < 0$, it has a finite upper endpoint (short/light tail).Since our focus is on heavy tails in wind extremes, we restrict attention to the case $\eta_{j,s} > 0$ (enforced by the positive parameterization introduced below).

**First-order pairwise likelihood decomposition.** Daily wind extremes often display short-range temporal dependence—consecutive days tend to co-vary. To address this, Fawcett & Walshaw (2006) model adjacent-day pairs and aggregate them via a first-order (Markov) composite likelihood. Within a fixed $(j, s)$ cell, write $x_t = X_{(j,s),t}$ for brevity, the composite likelihood in terms of $(\sigma_{j,s}, \eta_{j,s})$ is

$$\mathcal{L}_{j,s}(\sigma_{j,s}, \eta_{j,s}) = f(x_1 \mid \sigma_{j,s}, \eta_{j,s}) \prod_{t=1}^{n_{j,s}-1} \frac{f(x_t, x_{t+1} \mid \sigma_{j,s}, \eta_{j,s})}{f(x_t \mid \sigma_{j,s}, \eta_{j,s})}, \qquad (9)$$

where $f(x_t, x_{t+1} \mid \sigma_{j,s}, \eta_{j,s})$ is the joint density for the consecutive pair and $f(x_t \mid \sigma_{j,s}, \eta_{j,s})$ is the corresponding marginal. The joint density is specified in the next paragraph via a logistic bivariate extreme-value model after transforming each marginal to an extreme-value scale.

**Logistic bivariate extreme–value model after marginal transformation.** Fix a station–season cell and suppress indices. Let $u$ be the threshold, $\Lambda$ the exceedance rate, and $(\sigma > 0, \eta > 0)$ the GPD parameters. Transform $x > u$ to an extreme–value scale via

$$Z(x) = \Lambda^{-1} \left( 1 + \frac{\eta(x-u)}{\sigma} \right)^{1/\eta}.$$

The joint CDF for a consecutive pair $(x_t, x_{t+1})$ on this (generalized–Pareto) scale is

$$F(x_t, x_{t+1} \mid \sigma, \eta, \alpha) = 1 - \left[ Z(x_t)^{-1/\alpha} + Z(x_{t+1})^{-1/\alpha} \right]_+^{\alpha}, \qquad \alpha \in (0, 1], \qquad (10)$$

where $(a)_+ = \max(a, 0)$. Here, $\alpha = 1$ corresponds to independence, and $\alpha \to 0^+$ yields complete dependence.

**Region–wise contributions for consecutive pairs.** For each $(j, s)$ and consecutive pair $(x_t, x_{t+1})$, define the four regions by thresholding at $u_{j,s}$:

$$R_{11} : \{x_t > u,\ x_{t+1} > u\}, \quad R_{10} : \{x_t > u,\ x_{t+1} \le u\}, \quad R_{01} : \{x_t \le u,\ x_{t+1} > u\}, \quad R_{00} : \{x_t \le u,\ x_{t+1} \le u\},$$

with $u \equiv u_{j,s}$ and $x_t = X_{(j,s),t}$ for brevity. Let $Z(\cdot)$ be as above and write

$$\frac{\partial Z(x)}{\partial x} = \frac{Z(x)}{\sigma_{j,s}\big(1 + \eta_{j,s}(x - u)/\sigma_{j,s}\big)} \qquad (x > u).$$

Using $F(\cdot, \cdot \mid \sigma_{j,s}, \eta_{j,s}, \alpha_j)$ from equation 10, the numerator $f(x_t, x_{t+1} \mid \sigma_{j,s}, \eta_{j,s})$ in equation 9 is

**(i)** $R_{11}$ : $f(x_t, x_{t+1} \mid \sigma_{j,s}, \eta_{j,s}) = \dfrac{\partial^2 F}{\partial x_t\, \partial x_{t+1}}\Big|_{(x_t, x_{t+1})} = \dfrac{\partial^2 F}{\partial z_1\, \partial z_2}\Big|_{(Z(x_t), Z(x_{(t+1)}))} \dfrac{\partial Z(x_t)}{\partial x} \dfrac{\partial Z(x_{(t+1)})}{\partial x},$

**(ii)** $R_{10}$ : $f(x_t, x_{t+1} \mid \sigma_{j,s}, \eta_{j,s}) = \dfrac{\partial F}{\partial x_t}\Big|_{(x_t,\, u^+)} = \dfrac{\partial F}{\partial z_1}\Big|_{(Z(x_t),\, Z(u^+))} \dfrac{\partial Z(x_t)}{\partial x},$

**(iii)** $R_{01}$ : $f(x_t, x_{t+1} \mid \sigma_{j,s}, \eta_{j,s}) = \dfrac{\partial F}{\partial x_{t+1}}\Big|_{(u^+,\, x_{t+1})} = \dfrac{\partial F}{\partial z_2}\Big|_{(Z(u^+),\, Z(x_{t+1}))} \dfrac{\partial Z(x_{(t+1)})}{\partial x},$

**(iv)** $R_{00}$ : $f(x_t, x_{t+1} \mid \sigma_{j,s}, \eta_{j,s}) = F(u^+, u^+ \mid \sigma_{j,s}, \eta_{j,s}, \alpha_j) = 1 - \Big[ Z(u^+)^{-1/\alpha_j} + Z(u^+)^{-1/\alpha_j} \Big]_+^{\alpha_j},$

where $u^+$ denotes evaluation at the threshold from the exceedance side. The denominator $f(x_t \mid \sigma_{j,s}, \eta_{j,s})$ in equation 9 equals the GPD density $h(x_t - u \mid \sigma_{j,s}, \eta_{j,s})$ if $x_t > u$, and the non–exceedance mass otherwise.

**Overall likelihood.** The full composite likelihood is $\mathcal{L}(\theta) = \prod_{j=1}^{J} \prod_{s=1}^{S} \mathcal{L}_{j,s}(\theta)$ with $\mathcal{L}_{j,s}(\theta)$ as in equation 9 and region-wise $c_{j,s,t}$ given above. All other implementation details (threshold choice, numerical derivatives at $u^+$, and season-specific handling of $\Lambda_{j,s}$) are deferred to the Appendix.

**Parameterization.** We dispense with hierarchical random effects and use a non–hierarchical positive parameterization. For $j \in \{1, \dots, 4\}$ and $s \in \{1, \dots, 4\}$,

$$\sigma_{j,s} = \text{softplus}\big(\gamma_j^{(\sigma)}\big) + \text{softplus}\big(\varepsilon_s^{(\sigma)}\big), \qquad \eta_{j,s} = \text{softplus}\big(\gamma_j^{(\eta)}\big) + \text{softplus}\big(\varepsilon_s^{(\eta)}\big),$$

which enforces $\sigma_{j,s} > 0$ and $\eta_{j,s} > 0$. This additive form models station and season effects separately. For extremal dependence, each station has its own parameter $\alpha_j \in (0, 1)$. Since the normalizing flows operate most naturally on unconstrained real supports, we introduce $a_j^* \in \mathbb{R}$ with $\alpha_j = \text{sigmoid}(a_j^*)$ and perform inference on $a_j^*$.

**Prior and experiment settings.** We assign independent priors to the 20 parameters as follows: the raw scale effects $\gamma_{1:4}^{(\sigma)}$ and $\varepsilon_{1:4}^{(\sigma)}$ receive Student-$t_{\nu=10}$, and the raw shape effects $\gamma_{1:4}^{(\eta)}$ and $\varepsilon_{1:4}^{(\eta)}$ receive Student-$t_{\nu=3}$. This choice allows the posterior to accommodate dimension-specific tail thickness (lighter tails for the $\sigma$-effects, heavier tails for the $\eta$-effects). For extremal dependence, $\alpha_j \in (0, 1)$ is given a $\text{Beta}(1, 1)$ prior; as noted above, we work with $a_j^* \in \mathbb{R}$ via $\alpha_j = \text{sigmoid}(a_j^*)$, and include the change-of-variables term $\sum_{j=1}^{4} \log\{\alpha_j(1 - \alpha_j)\}$ in the log-likelihood.

Baseline MCMC uses an adaptive random walk Metropolis sampler (Haario et al., 2001). The proposal covariance is updated online to follow the local posterior geometry, which improves mixing for correlated and moderately high-dimensional targets while requiring neither gradients nor extensive tuning. This sampler is widely used in applied Bayesian analysis—especially for hierarchical models, state-space time series, and latent-variable settings—and is available in mainstream software. The code is implemented in $\mathbb{R}$ and was run on a dual-socket Intel Xeon Silver 4510 system with 24 physical cores and 48 threads (peak 4.10 GHz) under x86_64 Linux.

Figure 6 compares the estimated posteriors across all methods. Among the flow-based approaches, StiCTAF most consistently recovers the correct tail thickness in the majority of dimensions and captures the overall density shape with high stability.

Table 5 reports the posterior modes and 99% equal-tailed credible intervals for each method, together with computing time. StiCTAF requires more training time than other flows because it optimizes parameters at the component level, yet it remains far faster than MCMC while achieving comparable accuracy. Notably, flows with heavy-tailed bases such as TAF, gTAF, and ATAF do not fully recover the 99% intervals, whereas StiCTAF delivers reliable tail behavior and outperforms all baselines.

Table 5: Posterior modes with 99% equal–tailed credible intervals for all twenty parameters of the 2024 KMA wind dataset. The compared methods include MCMC (reference), NF (Gaussian/Gaussian mixture), and TAF variants (TAF, gTAF, gTAF mixture, ATAF, StiCTAF). Computing time (hours) is reported per method.

| Parameter | MCMC | StickTAF | NF(Gaussian) | NF(Gaussian Mixture) |
|---|---|---|---|---|
| $\gamma_1^{(\sigma)}$ | 0.40 (-2.09, 1.66) | 0.38 (-1.96, 1.51) | 0.38 (-1.62, 1.58) | 0.37 (-1.82, 1.56) |
| $\gamma_2^{(\sigma)}$ | 1.46 (-0.43, 2.90) | 1.45 (-11.43, 2.72) | 1.47 (-0.43, 2.73) | 1.32 (-0.74, 2.78) |
| $\gamma_3^{(\sigma)}$ | 1.52 (-0.24, 2.91) | 1.49 (-0.75, 3.01) | 1.46 (-0.24, 2.72) | 1.45 (-0.28, 2.80) |
| $\gamma_4^{(\sigma)}$ | 0.40 (-1.87, 1.73) | 0.36 (-2.21, 1.57) | 0.39 (-1.80, 1.61) | 0.40 (-2.17, 1.63) |
| $\varepsilon_1^{(\sigma)}$ | 1.09 (-1.83, 2.55) | 0.99 (-0.99, 2.56) | 1.04 (-0.77, 2.45) | 1.15 (-0.82, 2.46) |
| $\varepsilon_2^{(\sigma)}$ | 0.29 (-2.27, 1.76) | 0.38 (-1.98, 1.78) | 0.38 (-1.70, 1.71) | 0.32 (-1.87, 1.80) |
| $\varepsilon_3^{(\sigma)}$ | 1.10 (-0.71, 2.85) | 1.32 (-0.65, 2.99) | 1.26 (-0.52, 2.74) | 1.14 (-0.46, 2.64) |
| $\varepsilon_4^{(\sigma)}$ | 0.56 (-1.60, 1.85) | 0.53 (-1.78, 1.88) | 0.55 (-1.48, 1.80) | 0.52 (-1.46, 1.92) |
| $\gamma_1^{(\eta)}$ | -2.35 (-16.55, -0.72) | -3.37 (-11.15, -0.70) | -2.45 (-6.50, -0.78) | -2.46 (-6.93, -0.71) |
| $\gamma_2^{(\eta)}$ | -1.17 (-9.62, 0.19) | -1.43 (-9.50, 0.30) | -1.23 (-4.89, 0.19) | -1.30 (-4.72, 0.22) |
| $\gamma_3^{(\eta)}$ | -1.96 (-10.06, -0.48) | -2.14 (-9.30, -0.50) | -2.04 (-5.42, -0.41) | -2.25 (-6.03, -0.40) |
| $\gamma_4^{(\eta)}$ | -1.67 (-11.47, -0.35) | -1.80 (-6.43, -0.18) | -1.78 (-5.02, -0.34) | -1.82 (-5.52, -0.44) |
| $\varepsilon_1^{(\eta)}$ | -1.69 (-11.21, -0.32) | -1.81 (-11.89, -0.27) | -1.72 (-5.89, -0.31) | -1.53 (-6.66, -0.40) |
| $\varepsilon_2^{(\eta)}$ | -2.02 (-11.95, -0.38) | -2.06 (-12.09, -0.51) | -1.95 (-6.29, -0.51) | -1.89 (-7.61, -0.51) |
| $\varepsilon_3^{(\eta)}$ | -1.52 (-9.18, -0.13) | -1.62 (-10.21, -0.26) | -1.50 (-4.71, -0.22) | -1.45 (-5.14, -0.14) |
| $\varepsilon_4^{(\eta)}$ | -2.09 (-11.64, -0.50) | -2.22 (-13.88, -0.71) | -2.14 (-5.98, -0.52) | -2.03 (-6.82, -0.45) |
| $\alpha_1^*$ | 0.50 (0.05, 1.02) | 0.58 (0.13, 1.06) | 0.52 (0.06, 1.02) | 0.55 (0.09, 1.07) |
| $\alpha_2^*$ | 0.81 (0.30, 1.33) | 0.78 (0.30, 1.32) | 0.77 (0.31, 1.30) | 0.79 (0.29, 1.32) |
| $\alpha_3^*$ | 0.72 (0.21, 1.23) | 0.72 (0.24, 1.27) | 0.72 (0.26, 1.23) | 0.67 (0.22, 1.24) |
| $\alpha_4^*$ | 0.55 (0.08, 1.08) | 0.60 (0.11, 1.13) | 0.53 (0.08, 1.04) | 0.56 (0.10, 1.07) |
| comp.time (hr) | 11.90 | 0.08 | 0.03 | 0.03 |

| Parameter | TAF | gTAF | gTAF Mixture | ATAF |
|---|---|---|---|---|
| $\gamma_1^{(\sigma)}$ | 0.38 (-1.78, 3.07) | 0.47 (-1.72, 1.63) | 0.43 (-1.86, 1.63) | 0.42 (-1.70, 1.73) |
| $\gamma_2^{(\sigma)}$ | 1.40 (-2.72, 2.98) | 1.41 (-0.36, 2.77) | 1.39 (-0.31, 2.72) | 1.42 (-0.08, 2.85) |
| $\gamma_3^{(\sigma)}$ | 1.49 (-0.89, 3.42) | 1.47 (-0.07, 2.86) | 1.49 (-0.25, 2.80) | 1.49 (-0.13, 2.93) |
| $\gamma_4^{(\sigma)}$ | 0.34 (-1.97, 3.23) | 0.45 (-1.91, 1.69) | 0.44 (-1.76, 1.66) | 0.35 (-1.63, 1.70) |
| $\varepsilon_1^{(\sigma)}$ | 0.98 (-3.44, 3.70) | 1.01 (-1.06, 2.54) | 1.08 (-0.87, 2.56) | 0.88 (-1.10, 2.47) |
| $\varepsilon_2^{(\sigma)}$ | 0.39 (-3.17, 3.44) | 0.34 (-2.02, 1.64) | 0.45 (-1.65, 1.70) | 0.44 (-2.36, 1.82) |
| $\varepsilon_3^{(\sigma)}$ | 1.14 (-2.96, 3.82) | 1.28 (-0.66, 2.88) | 1.17 (-0.70, 2.83) | 1.18 (-1.04, 2.82) |
| $\varepsilon_4^{(\sigma)}$ | 0.44 (-3.24, 4.10) | 0.66 (-1.48, 1.88) | 0.45 (-1.70, 1.78) | 0.39 (-2.25, 2.11) |
| $\gamma_1^{(\eta)}$ | -2.56 (-5.35, 0.61) | -2.41 (-9.71, -0.69) | -2.60 (-10.51, -0.76) | -2.47 (-7.15, -0.66) |
| $\gamma_2^{(\eta)}$ | -1.30 (-4.93, 3.82) | -1.15 (-6.67, 0.31) | -1.19 (-6.43, 0.26) | -1.2 (-4.33, 0.28) |
| $\gamma_3^{(\eta)}$ | -2.33 (-6.97, 0.84) | -1.87 (-7.66, -0.51) | -2.07 (-9.18, -0.46) | -2.02 (-6.07, -0.18) |
| $\gamma_4^{(\eta)}$ | -1.74 (-4.96, 2.63) | -1.68 (-7.56, -0.22) | -1.63 (-7.50, -0.31) | -1.64 (-5.45, -0.15) |
| $\varepsilon_1^{(\eta)}$ | -1.70 (-4.71, 0.05) | -1.66 (-6.73, -0.27) | -1.71 (-7.24, -0.34) | -1.79 (-6.15, -0.11) |
| $\varepsilon_2^{(\eta)}$ | -2.60 (-7.12, 1.89) | -2.17 (-8.20, -0.47) | -1.90 (-8.04, -0.45) | -2.04 (-7.91, -0.11) |
| $\varepsilon_3^{(\eta)}$ | -1.64 (-6.04, 1.02) | -1.56 (-6.54, -0.12) | -1.63 (-6.74, -0.22) | -1.45 (-5.85, 0.01) |
| $\varepsilon_4^{(\eta)}$ | -2.32 (-5.31, 0.59) | -1.96 (-8.82, -0.47) | -1.91 (-9.21, -0.45) | -2.16 (-6.62, -0.35) |
| $\alpha_1^*$ | 0.50 (-0.14, 1.23) | 0.48 (0.03, 0.97) | 0.50 (0.04, 0.99) | 0.53 (0.08, 1.00) |
| $\alpha_2^*$ | 0.79 (-0.60, 2.78) | 0.79 (0.33, 1.32) | 0.78 (0.31, 1.30) | 0.77 (0.26, 1.33) |
| $\alpha_3^*$ | 0.70 (-0.10, 2.11) | 0.70 (0.22, 1.22) | 0.74 (0.20, 1.24) | 0.72 (0.16, 1.26) |
| $\alpha_4^*$ | 0.55 (-0.25, 1.56) | 0.55 (0.08, 1.08) | 0.54 (0.08, 1.07) | 0.50 (-0.03, 1.04) |
| comp.time (hr) | 0.03 | 0.03 | 0.08 | 0.03 |

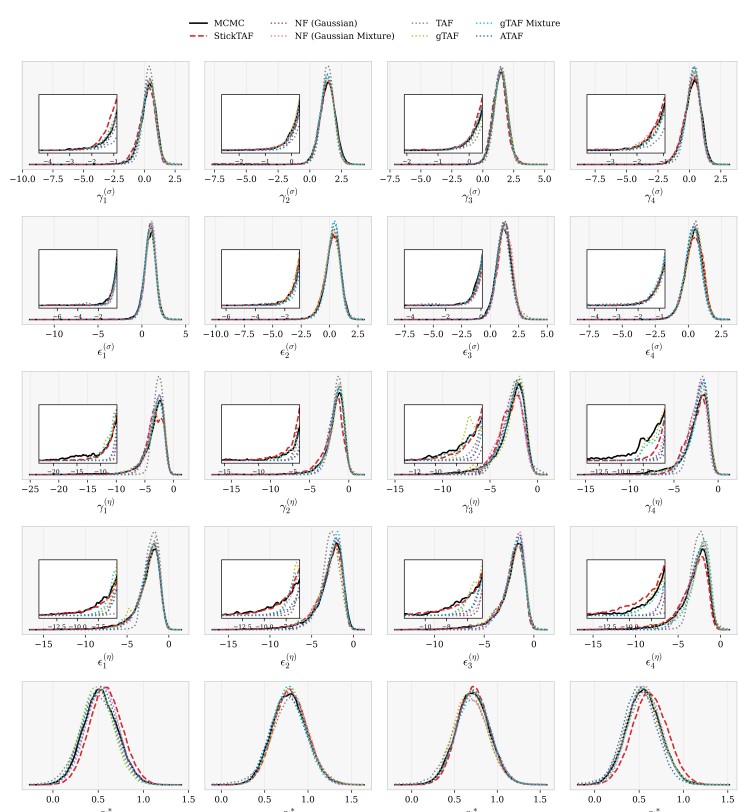

Figure 6: **Estimated posteriors for all twenty parameters from the real-data analysis.** Insets highlight the left 5% tail. Black curves show the MCMC reference; red curves show StiCTAF.

## C    DETAILS OF GTAF-MIXTURE

This appendix provides the essential details of gTAF-mixture, our extension of gTAF (Laszkiewicz et al., 2022), which augments the method with a stick-breaking mixture of Student-$t$ bases instead of a single Student-$t$ base.

### C.1    REPARAMETERIZED TRUNCATED STICK-BREAKING WEIGHTS

For a mixture of $J$ components, we use a truncated stick-breaking prior with concentration parameter $\alpha > 0$:

$$v_k \sim \text{Beta}(1, \alpha), \quad w_k = v_k \prod_{j=1}^{k-1}(1 - v_j), \ (k = 1, \dots, J-1), \qquad w_J = \prod_{j=1}^{J-1}(1 - v_j).$$

To enable pathwise gradients, we employ the inverse-CDF reparameterization for $\text{Beta}(1, \alpha)$:

$$v_k = 1 - \epsilon_k^{1/\alpha}, \qquad \epsilon_k \sim \mathcal{U}(0, 1),$$

and then deterministically map $\{v_k\}$ into $\{w_k\}$ as above. We deliberately choose this variant because it has a simple closed-form reparameterization, allowing pathwise gradients without introducing additional approximations or custom samplers that a more general Beta reparameterization would require.

## C.2 TAIL-INDEX PROFILING

**Pilot base and center pushforward.** We train a mixture of diagonal Gaussians with stick-breaking weights $\{w_1, \ldots, w_J\}$, whose density is given by

$$q_0(z) = \sum_{j=1}^{J} w_j \, \mathcal{N}\big(z \mid \mu_j, \operatorname{diag}(\sigma_0^2)\big), \qquad \sigma_0^2 \in (0, \infty).$$

We then push each center through a normalizing flow model $f$ at its initial state to compute

$$x_j = f(\mu_j) \in \mathbb{R}^D.$$

We retain only the components with sufficiently large weights:

$$\mathcal{J}_{\text{valid}} = \big\{ j \in \{1, \ldots, J\} : w_j \geq w_{\min} \big\}, \qquad w_{\min} = 0.1.$$

**Radius clustering and representatives.** Within $\{x_j\}_{j \in \mathcal{J}_{\text{valid}}}$ we perform fixed-radius clustering with radius $r = \rho\sqrt{D}$, where $\rho > 0$. From each cluster, we retain the member with the largest $w_j$. Let $\mathcal{M} \subseteq \mathcal{J}_{\text{valid}}$ denote the selected indices and $\{x_m\}_{m \in \mathcal{M}}$ their centers.

**Local linearization and anchors.** For each $m \in \mathcal{M}$, we approximate the pushforward covariance at $z = \mu_m$ using the Jacobian

$$J_m = \left. \frac{\partial f(z)}{\partial z} \right|_{z = \mu_m} \in \mathbb{R}^{D \times D}, \qquad \Sigma_{x,m} \approx J_m \operatorname{diag}(\sigma_0^2) \, J_m^\top.$$

For per-dimension and per-component standard deviations $\sigma_{x,m,i} = \sqrt{(\Sigma_{x,m})_{ii}}$, we define the dimension-wise standard deviation as

$$\sigma_i = \max_{m \in \mathcal{M}} \sigma_{x,m,i}, \qquad i = 1, \ldots, D.$$

We also record per-dimension anchors from the selected centers:

$$a_i^R = \max_{m \in \mathcal{M}} (x_m)_i, \qquad a_i^L = \min_{m \in \mathcal{M}} (x_m)_i,$$

and collect the anchor vectors $\mathbf{a}^R = (a_1^R, \ldots, a_D^R)$ and $\mathbf{a}^L = (a_1^L, \ldots, a_D^L)$.

**Log-log slope proxies.** Following Section 3.2, for each coordinate $i \in \{1, \ldots, D\}$ we draw i.i.d. samples $\{t_1^{(i)}, \ldots, t_n^{(i)}\}$ from a low–degrees-of-freedom Student-$t$ distribution and form the order statistics $t_{(1)}^{(i)}, \ldots, t_{(n)}^{(i)}$. Using the right-anchor vector $\mathbf{a}^R$ together with the scale $\sigma_i$, we obtain the right-tail estimate $\widehat{\nu}_i^R$. Analogously, for the left tail we use the left-anchor vector $\mathbf{a}^L$, scale $\sigma_i$, and order statistics to obtain $\widehat{\nu}_i^L$.

We then combine both sides while ensuring that the estimated degrees of freedom exceed 2:

$$\widehat{\nu}_i = \max\Big\{ \nu_{\min}, \, \min\big(\widehat{\nu}_i^L, \widehat{\nu}_i^R\big) \Big\}, \qquad \nu_{\min} = 2,$$

and cap overly light estimates:

$$\widehat{\nu}_i \leftarrow \min\{\widehat{\nu}_i, \, \nu_{\text{light}}\}, \qquad \nu_{\text{light}} = 30.$$

Finally, using $\{\widehat{\nu}_i\}_{i=1}^D$, we partition coordinates into

$$L = \big\{ i : \widehat{\nu}_i \geq \nu_{\text{light}} \big\}, \qquad H = \big\{ i : \widehat{\nu}_i < \nu_{\text{light}} \big\},$$

and reorder them so that all indices in $L$ precede those in $H$. This partition and permutation are used in the main construction: the light-tailed marginals share a common degrees-of-freedom parameter, while the heavy-tailed marginals retain their per-dimension initial values $\widehat{\nu}_i$ for subsequent learning.

## C.3 CONSTRUCTION OF BASE DISTRIBUTION AND NORMALIZING FLOWS

**Base construction.**  For the base $q_0$, we use a stick-breaking mixture of product Student-$t$ distributions. For each $k$-th component, the coordinates factorize, with each marginal given by a Student-$t$ with per-dimension scale $\sigma_i > 0$ (shared across components) and degrees of freedom $\nu_i$ (initialized from Section C.2). The base density is

$$q_{0,k}(z) = \prod_{i=1}^{D} t_{\nu_i}\left(z_i \mid \mu_{ik}, \sigma_i^2\right), \qquad q_0(z) = \sum_{k=1}^{K} \pi_k\, q_{0,k}(z), \quad \sum_{k=1}^{K} \pi_k = 1, \ \ \pi_k > 0.$$

The stick-breaking weights follow

$$v_k \sim \mathrm{Beta}(1, \alpha), \qquad \pi_k = v_k \prod_{j=1}^{k-1}(1 - v_j), \ (k = 1, \dots, K-1), \qquad \pi_K = \prod_{j=1}^{K-1}(1 - v_j).$$

When initializing location parameters $\mu_k \in \mathbb{R}^D$, we select $K$ candidate points from an open ball in $\mathbb{R}^D$ centered at the origin (including the origin), denoted $\{x^{(m)}\}_{m=1}^{K}$. Each candidate is mapped to the base space via the initial inverse flow:

$$z^{(m)} = f^{-1}\left(x^{(m)}\right),$$

and ranked according to the target likelihood. The mixture centers are then set by assigning the top-ranked $\{z^{(m)}\}$ to $\{\mu_k\}_{k=1}^{K}$ in descending order.

For reference, the component log-density is

$$\log q_{0,k}(z) = \sum_{i=1}^{D} \left[ \log \Gamma\left(\tfrac{\nu_i+1}{2}\right) - \log \Gamma\left(\tfrac{\nu_i}{2}\right) - \tfrac{1}{2}\log(\nu_i \pi) - \log \sigma_i - \tfrac{\nu_i+1}{2}\log\left(1 + \tfrac{(z_i - \mu_{ik})^2}{\nu_i \sigma_i^2}\right) \right].$$

**Normalizing flow structure.**  We follow the flow structure of Section B for each block, but adapt the group permutation in the LU-linear permutation layer to improve heavy-tail learning. Using the previously defined groups $L$ (light) and $H$ (heavy), with $|L| = d_\ell$ and $|H| = D - d_\ell$, each block applies a block lower-triangular linear map

$$W = \begin{bmatrix} A & 0 \\ B & C \end{bmatrix}, \qquad A \in \mathbb{R}^{d_\ell \times d_\ell}, \ B \in \mathbb{R}^{(D-d_\ell) \times d_\ell}, \ C \in \mathbb{R}^{(D-d_\ell) \times (D-d_\ell)}.$$

