# OpenReview forum: "Stick-Breaking Mixture Normalizing Flows with Component-Wise Tail Adaptation for Variational Inference"
_ICLR.cc/2026/Conference — ICLR 2026 Conference Withdrawn Submission_

### Official Review · Reviewer_kTSR · 2025-10-20

**Soundness:** 3
**Presentation:** 3
**Contribution:** 3
**Rating:** 4
**Confidence:** 2

**Summary:**

The present paper tackles the issues of performing variational inference for posteriors that can be multimodal and feature heavy-tails using normalizing flows. This challenge is tackled by learning a mixture as a base distribution with a stochastic number of effective components (a stick-breaking mixture) that can adapt to the need of the problem. Additionally, a method is proposed to estimate tail indices in each direction of the cartesian basis for each of the components. These estimated indices are further used to transform each Gaussian component of the SBM through a Tail transform flow, a non-Lipschitz transformation slightly adapted from a previous paper to adjust the tail of the base.

The proposed method is tested on:
-  a 2-dimensional synthetic example where the posterior is over two independent parameters, one heavy-tailed and one light-tailed.
- A 2d synthetic mixture of both heavy-tailed and light-tailed components.
- A 3d, apparently unimodal real world example of a Bayesian inference task of wind speeds.

**Strengths:**

1. The paper tackles a timely topic with a rather simple and apparently efficient method.
2.  Explaining clearly the problem at end and previous attempts of the literature, the paper is mostly well written. (Still, suggestions below would easily improve the manuscript.)
3. The experiments discussed in the paper demonstrate the relevance of the method on low dimensional settings.

**Weaknesses:**

Main weaknesses:

4. The main weakness of the paper is limited numerical validation of the approach. Indeed, it would be easy to strengthen the benchmark by considering systematic toy experiments of increasing difficulty, using for instance the framework proposed in [1], where the dimension and the distance separating modes are systematically increased to test the limit of samplers on multimodal targets. In particular here, it is to be expected that the mode-seeking behavior of the reverse KL is more and more problematic as the distance between modes [2].
5. The paper does not discuss the limitation of the approach: scaling with dimensionality, computational cost of the approach and tradeoffs for instance in setting the maximum number of components K.

Presentation improvements:

6. It would be useful to add a paragraph putting back all the ingredients together. For instance, I was not sure if, after the TTF on each component, a traditional NF architecture (RealNVP or Spline) is added and learned as well. Making a final summary of all the learnable parameters would clarify the final proposition.
7. The real world example is difficult to follow. The parameters $\Lambda$, \$\eta$ and $\sigma$ are not introduced. What is the posterior distribution? Over which variables? Some details might be given in the appendix, but it would be preferable that the main text is more self-contained.
8. Using mixtures of flows, as done in the base here, to model multimodal distributions has been already proposed in several works that could be cited ([3] section VI, [4]).

Minor presentation:

9. The meaning of acronyms that are not widely known. ATAF, etc.. and in particular of the proposed method StiCTAF, should be given explicitly.
10. The notation $\beta$ appears to be overloaded in 4.1. Maybe writing for instance explicitly at bottom of page 6 $\beta$, $\sigma^2 \sim N(\mu, \sigma_0^2) \times {\rm Inv-Gamma}(a_0, b_0)$ would clarify.

- [1] Grenioux, Louis, Maxence Noble, and Marylou Gabrié. “Improving the Evaluation of Samplers on Multi-Modal Targets.” Paper presented at Frontiers in Probabilistic Inference: Learning meets Sampling. ICLR Workshop on Frontiers in Probabilistic Inference: Learning Meets Sampling, April 24, 2025. https://openreview.net/forum?id=d91E9RhVFU.
- [2] Soletskyi, Roman, Marylou Gabrié, and Bruno Loureiro. “A Theoretical Perspective on Mode Collapse in Variational Inference.” Machine Learning: Science and Technology 6, no. 2 (2025): 025056. https://doi.org/10.1088/2632-2153/adde2a.
- [3] Hackett, Daniel C., Chung-Chun Hsieh, Michael S. Albergo, et al. “Flow-Based Sampling for Multimodal Distributions in Lattice Field Theory.” arXiv:2107.00734. Preprint, arXiv, July 1, 2021. http://arxiv.org/abs/2107.00734.
- [4] Molina-Taborda, Ana, Pilar Cossio, Olga Lopez-Acevedo, and Marylou Gabrié. “Active Learning of Boltzmann Samplers and Potential Energies with Quantum Mechanical Accuracy.” Journal of Chemical Theory and Computation, ahead of print, American Chemical Society, October 6, 2024. https://doi.org/10.1021/acs.jctc.4c00506.

**Questions:**

11. How is the maximum number of components K chosen in practice? It would have been interesting to discuss this and what is the effective number learned in each of the numerical examples.
12. How is “j” chosen in the “top j” extremes used to estimate the tail index?
13. Can learning suffer from instabilities because of the underdetermined fitting problem? One mode of the model can for instance first target a few modes of the target and then collapse on one. Is there always another component to compensate? First papers trying to use mixtures as bases of NFs typically mentioned this type of problem of matching.
14. Have the authors also considered in their benchmark using a deterministic Gaussian mixture (instead of the SBM) and using the TTF to take care of the heavy tails? What would be the drawback and advantages of such an approach compared to their proposition?

---

### Official Review · Reviewer_LFAT · 2025-10-30

**Soundness:** 3
**Presentation:** 3
**Contribution:** 2
**Rating:** 4
**Confidence:** 4

**Summary:**

The authors improve accuracy of flow-based variational approximations, by considering an approximation that uses an infinite mixture as the base distribution. The method is analysed in terms of tail estimation and evaluated empirically in relatively simple and small-scale experiments.

**Strengths:**

The paper is well written and easy to understand. The idea is sound and the theoretical analysis that characterises the tail behaviour of the approximation makes it interesting, compared to merely demonstrating improved accuracy in general. The tail behaviour is nicely motivated by the real-world example in Section 5, which quantifies the accuracy specifically in terms of capturing the tails. Overall, this is a well executed paper with a clear message and no major flaws.

**Weaknesses:**

The main weakness is the limited scientific novelty, which in top-tier venues needs to be emphasised. The authors acknowledge that Gaussian mixtures have been used a base distributions before and the generalisation from a finite mixture to an 'infinite' one using a stick-breaking prior is relatively straightforward given Roeder et al. (2017). Moreover, the authors take the somewhat lazy approach of explicitly truncating the mixture, which makes it largely equivalent to a standard finite mixture in practice. Moreover, there is no discussion on how the truncation level is set. Tail behaviour of mixture-based flows has been analysed before as well, and the theoretical analysis for the specific variant here is similar to previous work.

The empirical evaluation is otherwise nice, but the problems are all fairly simple and the authors fail to demonstrate a case where the method would really be needed. I disagree that Section 4.2 would shown an example of "complex multimodal target" given that it is simply a four-component mixture in two dimensions. This is not a trivial target and the illustration is helpful as a demonstration, but also not a complex one by modern standards and not something that would convince me to specifically pick this approximation. The real data is well justified in light of the technical contribution, but it is also of a low dimensionality and an example where MCMC also works well. The proposed method is naturally orders of magnitude faster than MCMC, but MCMC is here still easily possible.

**Questions:**

Table 2 quantifies the tail accuracy using 99% credible intervals. How would the results look like if considering other intervals, like 95%? Is the difference limited to the extreme tails, or would we see an improvement already at a lower coverage level?

What did you use as the truncation level? Is the method sensitive to the choice? Can you quantitatively demonstrate improvement over finite mixtures of varying size?

---

### Official Review · Reviewer_X5da · 2025-10-30

**Soundness:** 3
**Presentation:** 3
**Contribution:** 3
**Rating:** 6
**Confidence:** 3

**Summary:**

The paper presents a normalizing flow architecture for VI with multi-modal and heavy tail distributions. The idea is to use a stick-breaking mixture model for the base distribution, adapt the tails of each component separately, and then transform each component using layers of the normalizing flow. The paper compares against various other NF architectures and presents results on synthetic two dimensional examples as well as one real data analysis.

**Strengths:**

The paper is well written and easy to follow. Different contributions are well motivated and succinctly described. The paper does a good job of breaking the problem of heavy tails and multimodality and addressing each of them separately- with stick breaking for multimodality and learning and transforming tail indexes for heavy tails. The results, while few, clearly demonstrate that the proposed architecture is better able to capture the tails and different modes than other architectures and the authors considered many different baselines.

**Weaknesses:**

I outline my few concerns here

- The examples given in the paper are for very small problems. Especially the synthetic examples are just two dimensional, so it is unclear how this scales to high dimensions. It would be instructive to see this scaling with dimensionality as that will determine the where this method is applicable.

- On the same note, there are no scaling studies shown with other parameters, for example tail indices. How does the performance decrease with increasing tail weights.

- The authors do not discuss computational complexity in detail. The authors show wallclock time against MCMC for one example, but it would be better to compare number of function (or gradient) calls (i.e., $\log p(x)$ or $\frac{d \log p(x)}{dx}$) as that is less sensitive to implementation details and can be dominating for more complex models. I think this would be higher for the authors than other NF approaches for the author as some calls are also made in fitting the tail index at every iteration.

- One thing I find a little concerning is the results for $\alpha$ parameter in the real world example. As shown in Fig. 6 in B.3, the histograms for the proposed method are biased and hence I suspect the  99%ile values for these will also be wrong. This is curious given that the distribution seems to be Gaussian and hence potentially easier to fit? The authors should comment on this.

**Questions:**

In addition to some points raised in the weakness, I have a few additional questions-

- While the components of the proposed method are well described, the end-to-end algorithm is less clear. It would be good to include an algorithm box and a summary section, describing what are all the variational parameters being fit and how is each iteration done if someone needs to implement it. It would be also good if (upon acceptance), authors share the code online.
- On the same note, it is unclear how many components are being fit with the SBM step in first stage, or is that also a vriational parameter being optimized over?

- For the normalizing flow part, unless I missed something, the discussion in the appendix suggests that every dimension is being transformed independently. If so, would this not limit the flexibility and why is it not possible to use coupling transforms.

- How many samples (z's and u's) are needed to do tail estimation and how sensitive is the quality of inference to this estimation? I suspect this will also get more difficult with increasing dimensions, so can the authors comment on that?

- The authors mention: "we consider a normalizing flow model with a stick-breaking heavy-tailed mixture base to demonstrate that a heavy-tailed mixture base alone is insufficient. " I seem to have missed where this is shown in the main-text. What is the label used for this in figures and tables?

- While I understand why authors show ESS and forward KL loss, it would still be useful to add a column for reverse KL/ELBO as that is the metric being optimized.

---

### Note · Authors · 2025-11-17

I have read and agree with the venue's withdrawal policy on behalf of myself and my co-authors.